# Learning from Preferences and Mixed Demonstrations in General Settings

## Abstract

Reinforcement learning is a general method for learning in sequential settings, but it can often be difficult to specify a good reward function when the task is complex. In these cases, preference feedback or expert demonstrations can be used instead. However, existing approaches utilising both together are either ad-hoc or rely on domain-specific properties. Building upon previous work, we develop a mathematical framework for learning from human data and based on this we introduce LEOPARD: Learning Estimated Objectives from Preferences And Ranked Demonstrations. LEOPARD can simultaneously learn from a broad range of data, including negative/failed demonstrations, to effectively learn reward functions in general domains. It does this by modelling the human feedback as reward-rational partial orderings over available trajectories. We find that when a limited amount of preference and demonstration feedback is available, LEOPARD outperforms baselines by a significant margin. Furthermore, we use LEOPARD to investigate learning from many types of feedback compared to just a single one, and find that a combination of feedback types is often beneficial.

## 1 Introduction

Reinforcement Learning (RL) is a branch of machine learning where an agent learns a behavioural policy by interacting with an environment and receiving rewards. These rewards are determined by a reward function that mathematically encodes the objective of the agent. For real-world practical applications of RL, such as robotics or Large Language Model (LLM) finetuning, the specification of the reward function poses a difficult challenge. Two popular RL subfields try to solve this problem by leveraging human data in order to learn what the reward function should be, typically by optimising a parameterised function such as a neural network.

Inverse RL (IRL) utilises human-provided demonstrations of the correct behaviour and tries to learn a reward function for which only the demonstrations, or similar behaviour, are near-optimal (Ng et al., 2000; Ziebart et al., 2008; Wulfmeier et al., 2015). RL from Human Feedback (RLHF) presents the human with pairs of agent–behaviour examples. For each pair, the human decides which piece of behaviour is better, and the reward function is trained to re-produce this preference (Christiano et al., 2017). Both methods iterate between reward model and agent training. For more details on IRL and RLHF, see Sections 2.1 and 2.2, respectively. For many applications it might be possible and desirable to generate and learn from both of these feedback types, rather than committing to a single one. The current standard approach is to first train on demonstrations and then finetune the resulting model with preferences (Ibarz et al., 2018; Palan et al., 2019; Bıyık et al., 2022). Some methods have been proposed to more effectively leverage the information encoded in both the preferences and demonstrations, but this is still largely ad-hoc or specific to certain domains (Krasheninnikov et al., 2021; Mehta & Losey, 2023; Brown et al., 2019). We discuss these methods further in Section 2.3.

Submitted to 39th Conference on Neural Information Processing Systems (NeurIPS 2025). Do not distribute.

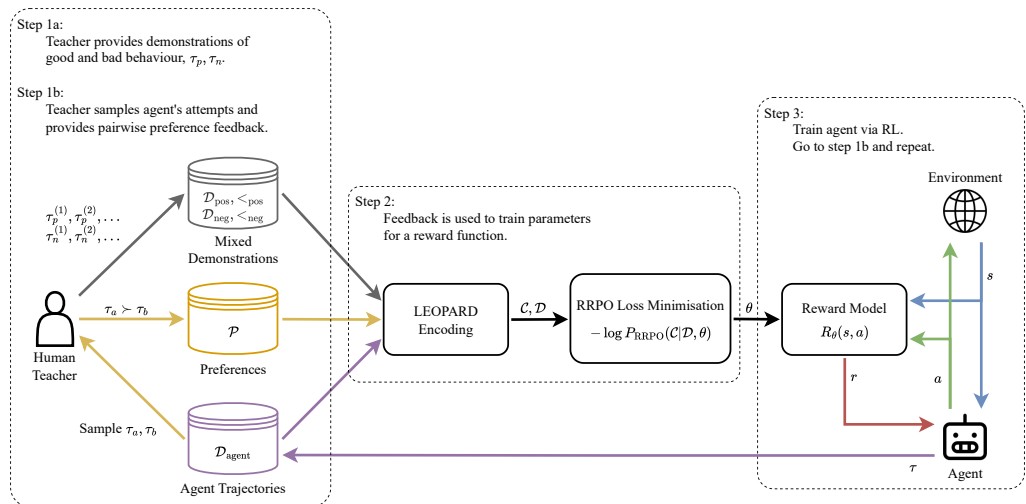

Figure 1: High-level overview of the LEOPARD algorithm. A teacher provides ranked examples of positive and negative demonstrations, as well as providing preference feedback over the agent's behaviour. This is used to train a reward model that the agent optimises via standard RL. The process is iterative. The LEOPARD encoding is given in Equations (6) and (7), and $P_{\text{RRPO}}$ is detailed in Equation (4).

In an attempt to solve this problem for general domains—and for many types of feedback including preferences and demonstrations—Jeon et al. (2020) propose Reward-Rational Choice (RRC). This frames the human feedback data as Boltzmann-Rational choices according to a probability distribution which has been induced by some unknown true reward function. Learning the reward function can then be cast as a supervised learning problem where we try to replicate these choices. Unfortunately, RRC is often difficult to implement in practice. For example, in the case of demonstration feedback, they treat it as a choice over all possible behaviours. This space is incredibly difficult to optimise over if it is very large and our reward function is non-linear, as is often the case for practical problems. Additionally, it cannot encode multiple selections for the 'optimal choice', nor can it encode more complex relationships between behaviours such as rankings or dis-preference.

To address these limitations, we introduce a new mathematical framework which frames the human feedback as *reward-rational partial orderings* over trajectories (RRPO). These partial orderings are then encoded by sets of Boltzmann-Rational choices, analogous to the Plackett-Luce ranking model (Marden, 1996). From this we derive LEOPARD: Learning Estimated Objectives from Preferences And Ranked Demonstrations, which is outlined in Figure 1. In addition to preferences and ranked (positive) demonstrations, LEOPARD can also learn from ranked negative/failed demonstrations. Preferences are interpreted as they are in RRC, but positive demonstrations are interpreted as being preferred to the agent's current and future behaviour, or the opposite in the case of negative demonstrations. Demonstration rankings, if available, are also cleanly translated into partial orderings.

LEOPARD can utilise a wide range of feedback types simultaneously, making it effective at learning useful reward functions in general environments. We find that when preference and positive demonstration feedback is available, it outperforms the standard baseline of performing DeepIRL on the demonstration data, and then finetuning using preferences. It also beats Adversarial Imitation Learning with Preferences (AILP), another preference and positive demonstration learning algorithm. Additionally, when only positive demonstration feedback is available, LEOPARD outperforms or matches DeepIRL and AILP due to its ability to exploit ranking data. Finally, we use LEOPARD to investigate learning from a variety of feedback types, compared to learning from a single one.

In summary, we make the following contributions:

1. We introduce RRPO, a practical and general framework for interpreting human feedback.

2. We introduce LEOPARD, an effective and scalable method for learning from preferences, and positive/negative ranked demonstrations.

3. We investigate learning from many types of feedback vs focussing on only a single one.

## 2 Related Work and Background

### 2.1 Demonstration-Based RL

A popular paradigm for learning from demonstrations is Inverse RL (IRL), where the demonstrations are used to learn a reward function (Ng et al., 2000). This overcomes many issues of behavioural cloning, which aims to directly mimic the given demonstrations (Bratko et al., 1995). Many current methods for IRL are based on the principle of *maximum (causal) entropy* (MaxEnt; MCE), established by Ziebart et al. (2008, 2010). This learns a reward function that captures the fact that the human demonstrations are optimal, but beyond this, it tries to have as much uncertainty about the reward dynamics as possible. Assuming a deterministic environment simplifies MCE into MaxEnt, and this assumption has been used to extend this class of methods into settings with high-dimensional observation spaces, e.g. DeepIRL (Wulfmeier et al., 2015). Advanced extensions of DeepIRL have been proposed, leveraging methods such as importance sampling (Finn et al., 2016), or GAN-style architectures (Fu et al., 2018). For a more comprehensive introduction to MCE and its derivatives, see Gleave & Toyer (2022). Our proposed algorithm does not reduce to a MaxEnt-derived method in the demonstration only case, but is still inspired by the principle and is of a similar form. Bayesian methods in IRL have also been explored (Ramachandran & Amir, 2007; Brown et al., 2020), highlighting how a probabilistic framing of the inverse learning problem can be useful.

### 2.2 Preference-Based RL

RLHF (Christiano et al., 2017) uses preferences—pairwise comparisons of agent behaviour—to learn a reward function for high-dimensional RL environments via the Bradley-Terry preference model (Bradley & Terry, 1952):

$$P_{\text{RLHF}}(\tau_a \succ \tau_b | \theta) = \frac{\exp(R_\theta(\tau_a))}{\exp(R_\theta(\tau_a)) + \exp(R_\theta(\tau_b))}, \tag{1}$$

where $R_\theta$ is a parameterised reward function and $\tau_a$ and $\tau_b$ are trajectory-fragments[1]. A 3-step iterative procedure is used: sampling of new comparisons of recent agent behaviour, fitting the reward model to the comparison dataset, and training of the policy on the learnt reward function. The reward model is fitted by minimising the average negative log-likelihood of the preference model across all pairs of trajectory-fragments. Wirth et al. (2017) provides a survey of other preference based RL methods prior to RLHF.

Recently, RLHF has been used for instruction and safety-finetuning large language models (LLMs) into chat systems (Ouyang et al., 2022; Bai et al., 2022; Bahrini et al., 2023). These are referred to as 'PPO-based' to disambiguate them from other methods which finetune LLMs from preferences without learning a reward function, such as DPO (Rafailov et al., 2024). Often the LLM is trained on demonstrations via behavioural cloning before PPO/DPO. Concerns for the safety, reliability, and misuse of LLMs has led to a plethora of research on how best to utilise human preferences/rankings to train these models (Cao et al., 2024; Chaudhari et al., 2024). Despite this, there is a broad lack of principled use of other feedback types for LLM safety and finetuning.

### 2.3 Combining Demonstrations and Preference Feedback

As mentioned in the case for LLMs, demonstration and preference feedback are typically combined by pre-training on the demonstration data using IRL/behavioural-cloning methods, and then finetuning the resulting reward model on preferences using RLHF (Ibarz et al., 2018; Palan et al., 2019; Bıyık et al., 2022). This works well in practice, but it is unclear how to add in further reward information, such as negative demonstrations or the relative rankings of demonstrations. Additionally, information that is present only in the demonstrations might be forgotten or never used, especially if strong regularisation is applied to the reward model, or the RL policy does not sufficiently explore when training on the demonstrations.

---

[1]Contiguous subsequences of trajectories.

More sophisticated combinations of preferences and demonstrations have been considered. Krasheninnikov et al. (2021) sampled trajectories according to reward functions optimal for the preferences, and applied MCE-IRL. This approach is computationally expensive and limited to linear reward functions over tabular MDPs. Mehta & Losey (2023) combine preferences and demonstrations alongside corrections (Bajcsy et al., 2017), but leverage domain-specific properties of robotics and encode their demonstrations using trajectory-space perturbations. This method is not applicable outside of robotics, and loses information about how demonstrations are better than most of trajectory-space, not just better than nearby trajectories. Brown et al. (2019) and Brown & Niekum (2019) both subsample ranked demonstrations to produce preferences for training the reward model, giving good results but still losing information about how those demonstrations might be preferred to other trajectories. Taranovic et al. (2022) combines a novel preference loss with adversarial imitation learning. This is the closest to our work, and so we test against it as a baseline. We also note that none of these methods can be easily extended to other types of feedback.

## 2.4 Learning From Other Types of Feedback

Other types of feedback have been explored in isolation, such as negative demonstrations (Xie et al., 2019),[2] improvements (Jain et al., 2015), off-signals (Hadfield-Menell et al., 2017a), natural language (Matuszek et al., 2012), proxy reward functions (Hadfield-Menell et al., 2017b), rankings (Myers et al., 2022), scalar feedback (Knox & Stone, 2008; Wilde et al., 2021), and even the initial state (Shah et al., 2019). Of these, Myers et al. (2022) is most similar to our work, as they use a Plackett-Luce model to to interpret rankings to train a reward model. We differ by considering many more types of feedback, showing how they can also be interpreted as orderings, and then use this to learn from preferences and mixed demonstrations.

Jeon et al. (2020) interpret many of types of feedback as part of an overarching formalism, *reward-rational (implicit) choice* (RRC), providing a mathematical theory for reward learning that combines different types of feedback. RRC interprets each piece of human feedback as a Boltzmann-Rational choice $C$ from some (possibly implicit) set of choices $\mathcal{D}$ with rationality coefficient $\beta$. A grounding function, $\psi$, maps choices to distributions over trajectories. The expected reward over these distributions gives the value for each choice under the Boltzmann-Rational model, according to some reward function $R_\theta$. For a deterministic $\psi$ simplifies to:

$$P_{\text{RRC}}(C|\mathcal{D}, \theta) = \frac{\exp(\beta R_\theta(\psi(C)))}{\sum_{C' \in \mathcal{D}} \exp(\beta R_\theta(\psi(C')))}. \tag{2}$$

Many of the formalisms of feedback in RRC, such as demonstrations, are not generally directly applicable as they naively require a large—possibly infinite—set of choices. Practical applications may rely on finite state-spaces, linear reward functions, unbounded surrogate loss functions, or sampling-based methods, each with their own pros and cons. We take inspiration from RRC, but show that formulating feedback as orderings leads to some more natural interpretations for mixed demonstrations without the need for such additional methods.

## 3 Method

We propose LEOPARD, a method for learning from preferences, positive demonstrations, negative demonstrations, and partial rankings over the given demonstrations. It is practical, flexible, and applicable to many environments. The aim is that a practitioner can give any and all feedback possible to the learning algorithm, and this feedback can be continuously learnt from and added to. First, we develop a general mathematical framework, reward-rational partial ordering (RRPO), extending that of deterministic reward-rational choice (RRC, Jeon et al. (2020)). Then, we apply this to the specific case of learning from preferences and mixed demonstrations.

### 3.1 Reward Rational Partial Orderings

To ensure the general applicability of our theoretical formalisms, we assume that only the trajectories our reward optimisation procedure has access to are provided directly. These could be generated

---

[2]They refer to these as 'failed demonstrations'.

during the agent's training or provided by the human in the case of demonstrations. This is assumed as sensible/relevant trajectories could sit on an unknown manifold in (a high-dimensional) observation space, crippling random-sampling based approaches.[3] We'd expect that reward functions capturing complex desirable behaviour would not be linear, but that they could at least be approximated sufficiently by some differentiable parameterised function.

Our key insight is to interpret human feedback as a set of Boltzmann-Rational choices encoding strict partial orderings over the trajectory-fragments we have direct access to, where a fragment is a contiguous subsequence of a trajectory. For each item in the partial order, we 'choose' that element out of a set containing itself and all elements strictly less than it. This is analogous to the Plackett-Luce ranking model (Marden, 1996), and is equivalent when the ordering can be viewed as a total ordering embedded in some larger set. Similar to RRC, each partial ordering is assumed to be independent given the reward function. Since a partial order may encode a single element being greater than all others with no other relations, this generalises deterministic choices of RRC.

Formally, let $\mathcal{D} = \{\tau_i\}_i$ be the set of all possible fragments of trajectories we have access to, $\mathcal{C} = \{<_j\}_j$ the set of human feedback, and $R_\theta$ our non-linear reward function parameterised by $\theta$. Note that $<_i$ is used to denote some partial ordering $i$. We define the likelihood of $\theta$ under RRPO as follows:

$$P_{\text{RRPO}}(\mathcal{C}|\mathcal{D}, \theta) = \prod_{(\tau_i, <_j) \in \mathcal{D} \times \mathcal{C}} P(<_j | \tau_i), \tag{3}$$

$$P(<_j | \tau_i) = \frac{e^{\beta_j R_\theta(\tau_i)}}{e^{\beta_j R_\theta(\tau_i)} + \sum_{\tau_k \in \mathcal{D}} \mathbf{1}_{\tau_k <_j \tau_i} e^{\beta_j R_\theta(\tau_k)}}, \tag{4}$$

where $\beta_j$ is the rationality coefficient for feedback $j$. $\beta$s should be equal if the type and source of feedback is the same, e.g. two pairwise preferences given by the same person. Note that when the partial orderings are sparse, many terms of the product become unity and can be ignored. We perform gradient descent on the negative-log of Equation (4) combined with a regularising term, giving the loss function below:

$$\mathcal{L}_{\text{RRPO}}(\theta) = -\log P_{\text{RRPO}}(\mathcal{C}|\mathcal{D}, \theta) + \mathcal{L}_{\text{Smooth}}(\mathcal{D}, \theta). \tag{5}$$

The smoothing term penalises the first derivative of the reward function over trajectories and leads to better shaped reward functions that are easier for the RL agent to learn from. It is inspired by previous work (Finn et al., 2016), and empirically we found it works well. Specific details are given in Section A.1.3.

A nice property of $\mathcal{L}_{\text{RRPO}}$ is that when minimised it faithfully represents the partial orderings. More precisely, upper bounds on the loss give rise to lower bounds on all reward differences between fragments that are related by some partial ordering. This is stated formally and proved in Theorem D.1 of Appendix D. As a special case, if the loss is below $\log 2$ then all reward differences must have the correct sign, i.e. the reward function induces an ordering compatible with all the partial orderings.

## 3.2 LEOPARD

Whilst we can apply the framework above to many types of feedback, we now focus on the case of combining preferences with mixed demonstrations. By mixed demonstrations, we mean ones which may be positive, negative and, within these two groups, we may have access to the relative rankings of each demonstration.

A pairwise preference of $\tau_a \succ \tau_b$ is simply interpreted as a partial ordering with only $\tau_b < \tau_a$.[4] Positive demonstrations are interpreted as a single partial ordering that prefers all positive demonstrations to any agent trajectories and encodes the relative rankings of the positive demonstrations themselves. Negative demonstrations are interpreted likewise, but these partial orderings prefer agent trajectories over the negative demonstrations.

Formally, let $\mathcal{D}_{\text{pos}}, <_{\text{pos}}$, and $\mathcal{D}_{\text{neg}}, <_{\text{neg}}$ be the sets of trajectories and partial orderings encoding rankings from positive and negative demonstrations, respectively. Let $\mathcal{D}_{\text{agent}}$ be the set of trajectories

---

[3]For example, consider the space of all images vs ones which are plausible 3D scenes.

[4]By interpreting each preference as its own partial ordering, we avoid potential issues of symmetry and non-transitivity.

sampled from the agent's behaviour. Let $\mathcal{P} = \{(\tau_a, \tau_b)_i\}_i$ be the set of ordered pairs of trajectory-fragments in which the first is preferred, and $R_\theta$ our parameterised reward function. Then we optimise the loss function, Equation (5), with:[5]

$$\mathcal{C} = \{<_{\text{Demo}}\} \cup \mathcal{C}_{\text{Pref}}, \tag{6}$$

$$\mathcal{D} = \bigcup\{\mathcal{D}_{\text{pos}}, \mathcal{D}_{\text{neg}}, \mathcal{D}_{\text{agent}}, \mathcal{D}_{\text{pref}}\}, \tag{7}$$

where the demonstration and preference partial orderings are given by:

$$
\begin{aligned}
<_{\text{Demo}} = \; &<_{\text{pos}} \cup <_{\text{neg}} \cup \{\tau_n < \tau_a < \tau_p, \\
&\quad |(\tau_n, \tau_a, \tau_p) \in \mathcal{D}_{\text{neg}} \times \mathcal{D}_{\text{agent}} \times \mathcal{D}_{\text{pos}}\}, \\
\mathcal{C}_{\text{Pref}} = \; &\{\{\tau_b < \tau_a\}|(\tau_a, \tau_b) \in \mathcal{P}\}, \\
\mathcal{D}_{\text{pref}} = \; &\bigcup_{(\tau_a, \tau_b) \in \mathcal{P}} \{\tau_a, \tau_b\}.
\end{aligned}
$$

Like in the case for RLHF, our dependencies on agent behaviour means we need to iterate between sampling new preferences, optimising for Equation (5), and training the agent's policy.[6] Our algorithm is illustrated in Figure 1 and the full training procedure is given in Algorithm 1 in Appendix A, along with details on reward model training.

## 4 Experiments

### 4.1 Experimental Setup

We test our method on several environments against common baselines in order to evaluate its performance across a broad variety of domains. Additionally, we also vary the proportions and amounts of different types of feedback used for learning to investigate the effects of this on performance. In order to reduce the cost of testing our method and facilitate hyperparameter tuning with many repetitions, we synthetically generate preferences, demonstrations, and their rankings. We generate preferences by sampling using the sigmoid of the reward difference between the two fragments under comparison as the probability of preference. We generate demonstrations by training an agent on the ground truth reward function and then sampling its trajectories, with their ground truth reward determining their relative rankings. For further details, see Section A.2. For each combination of environment, algorithm, and amount of feedback, we run 16 random seeds and report the average results with 1-$\sigma$ standard error. Standard errors are computed via the typical method of dividing the empirical variance by the square root of the sample size.

We evaluate on four environments from the Gymnasium (Towers et al., 2024) test suite: Half Cheetah (MuJoCo), Cliff Walking (Toy Text), Lunar Lander (Box2D), and Ant (MuJoCo). This covers a range of continuous and discrete observation and action spaces, reward sparsities, and overall complexities. We require a finite horizon to reduce complications from the preference and demonstration learning, so some environments required modification. These and other environment details and hyperparameters are given in Appendix B.

In order to get a good number of preferences and demonstrations to test with, we see how many preferences or positive demonstrations LEOPARD needs to get good performance in the single feedback type case, and then use a normalised weighted combination of these.[7] This allows us to be confident there is enough feedback for learning, but not so much that it's too easy.

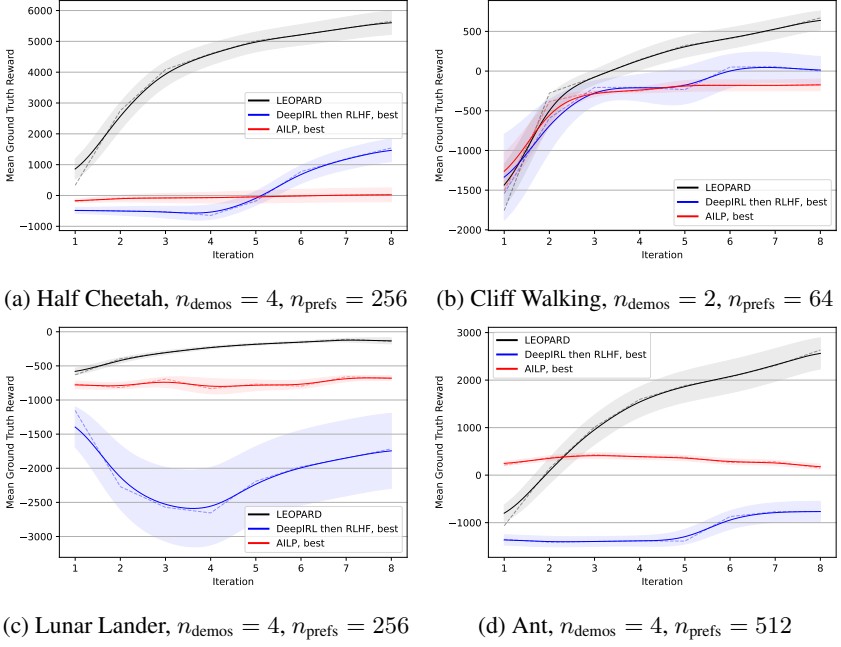

(a) Half Cheetah, $n_{\text{demos}} = 4$, $n_{\text{prefs}} = 256$    (b) Cliff Walking, $n_{\text{demos}} = 2$, $n_{\text{prefs}} = 64$

(c) Lunar Lander, $n_{\text{demos}} = 4$, $n_{\text{prefs}} = 256$    (d) Ant, $n_{\text{demos}} = 4$, $n_{\text{prefs}} = 512$

Figure 2: Comparison of LEOPARD with the baselines of AILP and DeepIRL followed by RLHF, when positive demonstrations and preferences are available. The lines denote the mean of the ground truth reward function, with shaded standard errors across 16 random seeds, against algorithm iterations—alternations between optimising the reward model and the agent. Solid lines are smoothed means for clarity, dashed lines give raw values. A breakdown of the performance of the baseline methods for different reward model training epochs per iteration is given in Figures 7 and 8.

## 4.2 LEOPARD vs Baselines

In Figure 2 we compare LEOPARD against Adversarial Imitation Learning with Preferences (AILP, Taranovic et al. (2022))[8] and a standard pipeline of training on demonstrations with DeepIRL and then preference finetuning with RLHF.

We see that without exception LEOPARD outperforms both baselines by a considerable margin. Since LEOPARD can utilise all the data all the time, preferences can be used to aid early exploration, and demonstrations can continue to be trained against even in the latter stages. Rankings over demonstrations provide an additional information source the baselines are unable to make use of. Additionally, as it trains the reward model to rough convergence each iteration it allows for adequate learning without over-fitting, and does not require tuning a 'reward model training epochs' hyperparameter.

When training the reward model with LEOPARD, we keep training until the loss has loosely converged (see Section A.1.2 for details). This is not possible with DeepIRL as the maximum-entropy 'loss' function is not bounded from below, thus the number of training epochs for the reward model is fixed. We try a variety of values and compare against the best, for a full breakdown see Figure 7. For AILP,

---

[5]An exception to this formulation is used for Cliffwalking, where agent trajectories can easily be as bad as negative demonstrations. The demo partial rankings are in this case split, one preferring positive demonstrations to agent trajectories, and another preferring positive to negative demonstrations.

[6]If there were an existing set of preferences and agent trajectories, the method could be applied offline by simply optimising for Equation (5).

[7]E.g. 50% of the maximum number of preferences + 50% of the maximum number of positive demonstrations.

[8]For our implementation of AILP we only use the relevant loss functions and disregard the extraneous parts of the method, namely initially optimising the policy to maximise visited state entropy and sampling preferences according to maximum entropy. We use the same RL algorithms as LEOPARD uses, as detailed in Appendix A. Overall this enables a fair comparison with LEOPARD, and we note that AILP's additional tweaks could be symmetrically applied to LEOPARD if desired.

we try using both our dynamic stopping and a fixed number of training epochs again comparing against the best, see Figure 8 in Appendix C for a breakdown of these results.

Whilst not the focus of our algorithm, we additionally show that with only positive demonstrations LEOPARD either beats or performs similarly to the baselines. This is shown in Appendix C, Figure 6, with the breakdowns of DeepIRL and AILP's results for different numbers of training epochs given in Figures 9 and 10 respectively.

Table 2 in Appendix C gives a numerical breakdown of final scores for each algorithm in each environment, including the different settings of AILP and DeepIRL.

Note that for the analysis of the Cliff Walking environment, outliers have been removed These were due to excessively large negative rewards from walking off the cliff many times before learning this was bad. A detailed breakdown is given in Appendix C, Table 4.

## 4.3 Learning from a Mixture of Feedback Types

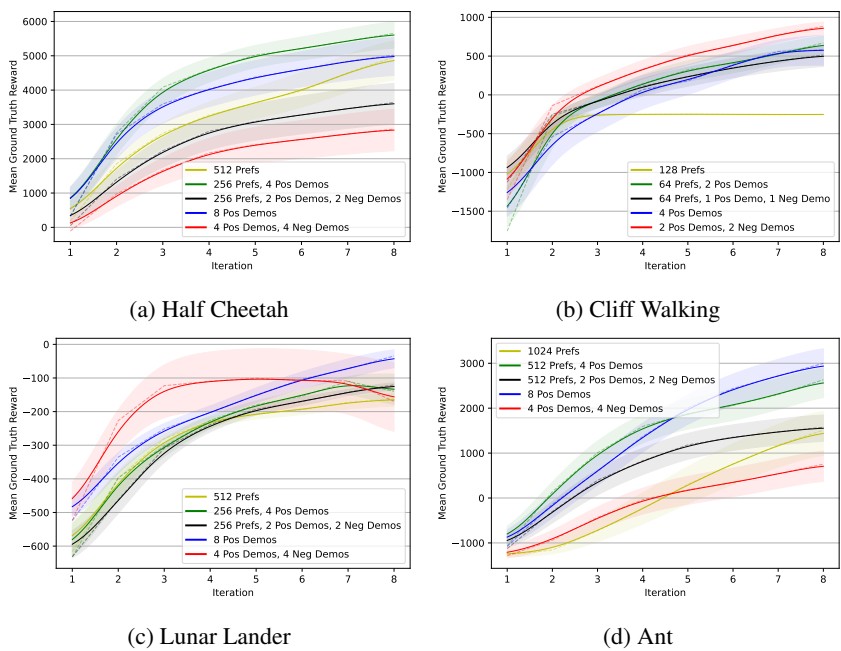

Figure 3: Comparison of LEOPARD's performance when varying types of feedback are available. The lines denote the mean of the ground truth reward function, with shaded standard errors across 16 random seeds, against algorithm iterations—alternations between optimising the reward model and the agent. Solid lines are smoothed means for clarity, dashed lines give raw values.

In Figure 3 we investigate the performance of LEOPARD when learning from a variety of different feedback proportions. Final scores are detailed in Appendix C, Table 3. The results are somewhat mixed and noisy, but we see that preferences combined with positive demonstrations consistently performs well.

## 5 Discussion

### 5.1 Generality of RRPO

Reward-rational preference orderings, the basis of LEOPARD, are a generalisation of the deterministic reward-rational choice framework (Jeon et al., 2020), but offers several distinct advantages. Recall that RRC frames the human feedback as a choice over some set, and then maps elements of that set into distributions over trajectories. Instead, RRPO maps the human feedback directly into a set of partial orderings. These two approaches have differing flexibility, and different feedback types might lend themselves more readily to one or the other. However, as RRPO is explicit in its construction

that it operates only over directly-accessible trajectories, it becomes much more general in a practical sense.

Furthermore, RRPO does not assume any particular properties about the space of reward functions, nor the space of trajectories. In general, one can think of optimal trajectories as a small part of some feasible-trajectory manifold, which itself is a small part in a larger trajectory feature space. Methods which rely on domain-specific properties of these spaces, such as linearity or computable perturbations, inherently limit themselves from being more broadly applied. For example, Mehta & Losey (2023) leverages inverse kinematics models to interpret demonstration feedback (alongside preferences) in robotics domains. Whilst effective for this application, it renders the broader method impossible outside of robotics. RRPO and LEOPARD on the other hand, could be easily applied to environments very different to the ones we have tested on. For example, they could be used for Large Language Model (LLM) finetuning.

## 5.2 Limitations and Future Work

Whilst we have tested LEOPARD on a range of environments with differently structured observation and action spaces, a more comprehensive study would investigate an even wider range of tasks, such as more complex robotics, Atari games, and even LLM finetuning. Furthermore, with additional resources, it would be instructive to more closely interrogate how performance depends on the proportions of different feedback used for learning. For instance, future work could vary the feedback proportions with greater precision and then fit and analyse simple predictive models on this.

Additionally, there are other methods that seek to learn from both preference and demonstration data, or even negative/failed demonstrations, as detailed in Sections 2.3 and 2.4. Whilst these are less general in application than LEOPARD; a comparison of performance would still be interesting. We have chosen the baselines of AILP and 'DeepIRL followed by RLHF' to test against as they have similar simplicity and generality to our own method, as well as the latter being common practice.

We introduce RRPO as a theoretical backdrop for LEOPARD, however our investigation of its properties and encodings for many types of feedback is limited. Due to its similarity to RRC and the Placket-Luce choice model, we do not see this as a critical failing, as it will inherit many properties from those models, and deterministic RRC formulations can be trivially encoded under RRPO. Nevertheless, there are likely important theoretical properties and applications of RRPO that are of relevance to reward learning that ought to be investigated.

## 6  Conclusion

We have shown that LEOPARD can perform effective reward inference, learning from many sources of reward information simultaneously. It is more effective than standard baselines for learning from preferences and demonstrations, and can additionally incorporate more information such as demonstration rankings and negative/failed demonstrations. We have also investigated how many sources of reward information could be more beneficial than relying on only large amounts of a single type. The generality and simplicity of our method makes it very powerful and applicable to important current problems such as high dimensional robotics, and LLM finetuning. Furthermore, it opens the door to exploring the use of a much wider range of feedback in many RL settings.

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

## A  Algorithm Details

The full algorithm for LEOPARD is given in Algorithm 1. Initialisations follow standard neural network initialisation methods. RandomRollouts generates trajectories by sampling random actions and resetting the environment when necessary. TrainAgent uses the standard SAC algorithm for when the action space is continuous, and PPO when it's discrete. For both algorithms we use the implementations provided by Stable Baselines3 (Raffin et al., 2021). It uses the learnt reward function to generate rewards for the RL procedure. Hyperparameters used for SAC and PPO are those given in RL Baselines3 Zoo (Raffin, 2020), except for Lunar Lander where we use an entropy bonus of 0.05 instead of 0. Details on TrainRewardModel and GetPreferences are given in Sections A.1 and A.2.1 respectively. The generation of the demonstrations and their rankings is detailed in Section A.2.2.

---

**Algorithm 1** LEOPARD

---

**Input:**

$n_{\text{iters}}$           Number of iterations to perform
$n_{\text{rollout-steps}}$      Number of environment rollout steps
$n_{\text{prefs}}$          Number of preferences to sample
$\mathcal{D}_{\text{pos}}$          Positive demonstrations
$<_{\text{pos}}$          Positive demonstrations partial ordering
$\mathcal{D}_{\text{neg}}$          Negative demonstrations
$<_{\text{neg}}$          Negative demonstrations partial ordering

**Output:**

$\pi$          Trained agent policy
$R_\theta$          Learnt reward function

$n_{\text{rollout-steps-per-iter}} \leftarrow \lfloor n_{\text{rollout-steps}}/(n_{\text{iters}} + 1) \rfloor$
$n_{\text{prefs-per-iter}} \leftarrow \lfloor n_{\text{prefs}}/n_{\text{iters}} \rfloor$
$\mathcal{D}_{\text{agent}} \leftarrow \emptyset$ {Agent trajectory pool}
$\mathcal{P} \leftarrow \emptyset$ {Preferences dataset}
$\pi \leftarrow \text{InitialiseAgent}()$
$R_\theta \leftarrow \text{InitialiseRewardFunction}()$
$\mathcal{D}_{\text{new-trajectories}} \leftarrow \text{RandomRollouts}(n_{\text{rollout-steps-per-iter}})$

**for** $i = 1$ **to** $n_{\text{iters}}$ **do**
    $\mathcal{P} \leftarrow \mathcal{P} \cup \text{GetPreferences}(n_{\text{prefs-per-iter}}, \mathcal{D}_{\text{new-trajectories}}, \mathcal{D}_{\text{agent}})$
    $\mathcal{D}_{\text{agent}} \leftarrow \mathcal{D}_{\text{agent}} \cup \mathcal{D}_{\text{new-trajectories}}$
    $R_\theta \leftarrow \text{TrainRewardModel}(R_\theta, \mathcal{D}_{\text{pos}}, <_{\text{pos}}, \mathcal{D}_{\text{neg}}, <_{\text{neg}}, \mathcal{D}_{\text{agent}}, \mathcal{P})$
    $\pi, \mathcal{D}_{\text{new-trajectories}} \leftarrow \text{TrainAgent}(\pi, R_\theta, n_{\text{rollout-steps-per-iter}})$
**end for**

---

### A.1  Reward Model Training

The reward model is trained by optimising the loss function Equation (5) with the AdamW optimiser. Batches of $\mathcal{D}_{\text{pos}}, \mathcal{D}_{\text{neg}}, \mathcal{D}_{\text{agent}}$, and $\mathcal{P}$ are sampled independently, and then encoded via Equations (6) and (7). Since we want to respect the relative proportions of each data source[9] but also have independent batch sizes, normalisation of the loss across the batch is slightly involved. This is detailed in Section A.1.1. Instead of training for a fixed number of steps / epochs, training steps are taken until a stopping condition is reached, as detailed in Section A.1.2. Together these procedures could result in varying coverages for each data source, from potentially many epochs on one,[10] to only sampling a small fraction of another.

---

[9]E.g. if we had 1000 preferences and 1 demonstration, we'd probably care more about low average loss from the preferences than from the demonstration.

[10]Since our data sources are of varying sizes and not partitioned into equal numbers of batches, the notion of a training epoch - one complete pass over all training data - is not well-defined. We do however have notions of data source specific epochs.

### A.1.1 Loss Normalisation Across Batch

As we want our gradient steps to be roughly unity in magnitude and independent of the batch size, we need to normalise it. Typically, this is very easy in supervised learning—one can simply take an average across the batch—but this is not the case for Equation (5). Expansion of the gradient of the loss with respect to $\theta$, and noting our reward function operates at the level of transitions within trajectories, reveals the normalising factor of each data source (note this assumes a fixed length of fragments for each partial ordering):

$$\sum_{(\tau_i, <_j) \in \mathcal{D} \times \mathcal{C}} \text{Length}(\tau_i) \cdot \mathbf{1}_{\exists \tau_k \in \mathcal{D}.\tau_k \neq \tau_i \wedge \tau_k <_j \tau_i}.$$

The loss term of each data source is first divided by this factor evaluated on the batch—so that they are all at most unity in magnitude—and then combined in a weighted sum where the weights are the factors evaluated on the whole dataset for that source divided by the sum of these dataset-level factors. Some data sources, namely $\mathcal{D}_{\text{agent}}$, are treated as 'in-excess', and their dataset-level factor is made proportional to another data source, e.g. $\mathcal{D}_{\text{pos}}$.

### A.1.2 Stopping Conditions

Generally, the reward function loss from poorly-fitted demonstration rankings are much higher than poorly fitted preferences. This is because trajectories are typically longer than trajectory-fragments and demonstrations generate more '$<$' comparisons than a preference. However, the distribution of demonstrations are typically quite far from that of the agent trajectories, which the preferences have been generated over. This makes it much easier for the reward function to separate the demonstrations from agent behaviour and thus achieve a low loss on the demonstration ordering, than it does for it to get low loss on all the preference orderings.

The consequence of the above two facts is that if we were training on just the demonstrations, we'd want to do at most a few epochs (to learn fast and avoid overfitting), but if we were training on just the preferences we might want to do more (as learning is slower and overfitting less of a potential issue). Thus, as the amount of data in each dataset varies in each iteration, it does not make sense to have a pre-specified number of training steps, and instead a stopping condition should be used.

Our stopping condition simply checks if the training loss has loosely converged. At each step we check if the change in training loss is less than 10% of the last step's training loss. If this occurs 3 times in a row, we stop training the reward model for that iteration, and return to agent training. There is a hard limit of 256 epochs on the smallest data source, though this is rarely reached. Empirically this strikes the balance between learning the most from the small amount of data, and avoiding overfitting.

### A.1.3 Smoothness Loss

In addition to our negative log-likelihood loss term for optimising RRPO, we also have a loss term based on the smoothness of the reward function over trajectories, as seen in Equation (5). This is defined as proportional to the mean-squared first derivative in reward with respect to environment step for all full trajectories.[11] Concretely:

$$\mathcal{L}_{\text{Smooth}}(\mathcal{D}, \theta) = \mu_{\text{smooth}} \frac{1}{|\mathcal{D}_{\text{Full}}|} \sum_{\tau_i^{(n)} \in \mathcal{D}_{\text{Full}}} \frac{1}{n-1} \sum_{k=1}^{n-1} (R_\theta(s_{k-1}, a_{k-1}, s_k) - R_\theta(s_k, a_k, s_{k+1}))^2,$$

$$\tag{8}$$

$$\mathcal{D}_{\text{Full}} = \{\tau_i | \tau_i \in \mathcal{D}, \forall \tau_{j \neq i} \in \mathcal{D} . \tau_i \not\subset \tau_j\}, \tag{9}$$

$$\tau_i^{(n)} = \{(s_0, a_0, s_1), ..., (s_{n-1}, a_{n-1}, s_n)\}. \tag{10}$$

We set $\mu_{\text{smooth}}$ to 0.1 based on early empirical results.

---

[11]I.e. not fragments used for preferences.

## A.2 Synthetic Feedback

### A.2.1 Preferences

In Algorithm 1, the GetPreferences function randomly samples trajectory fragments for comparison, with a bias to sampling from new trajectories. We are using a synthetic oracle which uses the ground truth reward function to noisily generate preferences, simulating the imperfect human rationality. More specifically, for each sampled pair of fragments, the sigmoid of their reward difference is used as the parameter for a Bernoulli random variable which is then sampled to generate the preference.

### A.2.2 Demonstrations

To create demonstrations for our tasks, we simply train an agent on the ground truth reward function (or its negation in the case of negative demonstrations). Several agents are trained, and the best few, $n_{\text{selected}}$, are picked. From these agents, we create a list of their trajectories, ordering from their latest attempts to their first, and interleaving each agent together with the best agent first. For training an agent from feedback, if $n$ demonstrations are being used, the first $n$ demonstrations from this list are provided. Rankings are generated automatically based on the ground truth reward of each demonstration, making $<_{\text{pos}}$ and $<_{\text{neg}}$ total orders.[12] The ground truth reward per agent step and number selected, $n_{\text{selected}}$, of all demonstrations trained are given in Figures 4 and 5 for positive and negative demonstrations respectively.

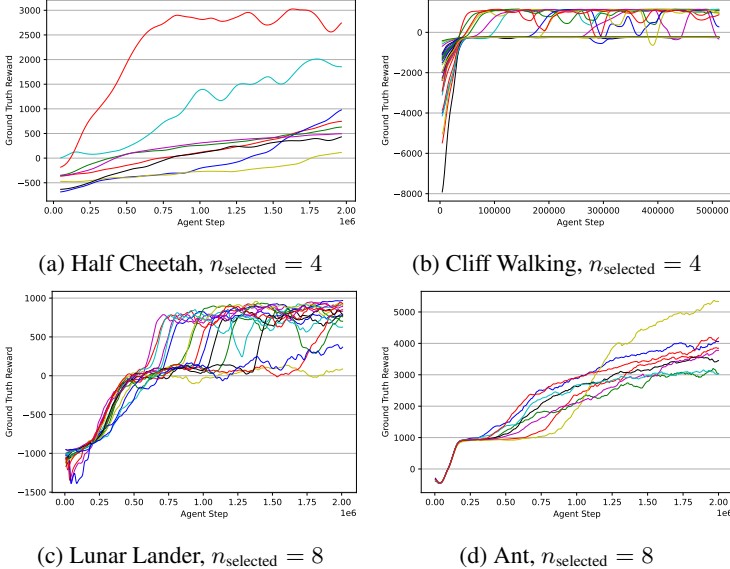

(a) Half Cheetah, $n_{\text{selected}} = 4$

(b) Cliff Walking, $n_{\text{selected}} = 4$

(c) Lunar Lander, $n_{\text{selected}} = 8$

(d) Ant, $n_{\text{selected}} = 8$

Figure 4: Ground truth reward vs agent steps for the positive demonstrations that were trained in every environment. We also state how many were selected as good examples to be used for demonstration learning.

---

[12]They are not required to be total orders to apply the general method.

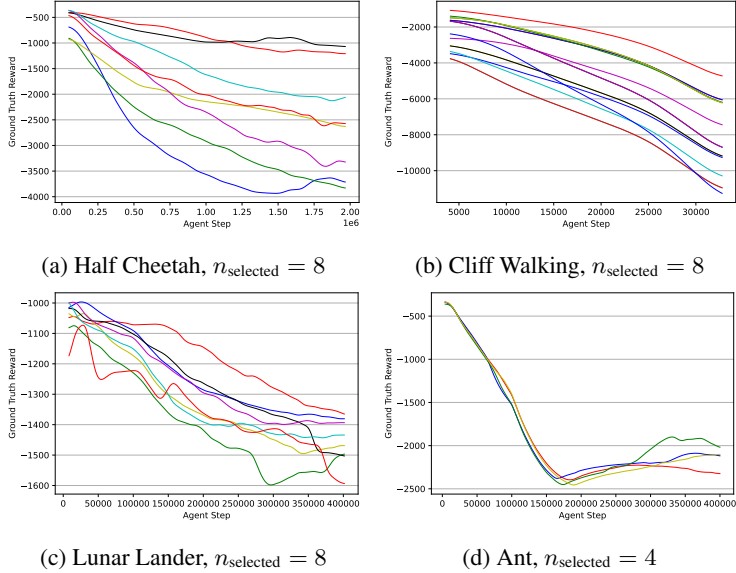

(a) Half Cheetah, $n_{\text{selected}} = 8$    (b) Cliff Walking, $n_{\text{selected}} = 8$

(c) Lunar Lander, $n_{\text{selected}} = 8$    (d) Ant, $n_{\text{selected}} = 4$

Figure 5: Ground truth reward vs agent steps for the negative demonstrations that were trained in every environment. We also state how many were selected as bad examples to be used for demonstration learning.

## B  Experiment and Environment Details

Here we give details on versions / modifications made for each environment, as well as environment-specific hyperparameters summarised in Table 1. We used $n_{\text{iters}} = 8$ and 16 random seeds for all runs.

Table 1: Environment specific hyperparameters. 'Trajectory Length' refers to the fixed time horizon for that environment, 'Preference Fragment Length' is the length of the contiguous trajectory subsequences that are used to generate preferences. Both are measured in environment timesteps.

| Environment | Trajectory Length | Preference Fragment Length | $n_{\text{rollout-steps}}$ |
|---|---|---|---|
| Half Cheetah | 1k | 32 | 2M |
| Cliff Walking | 250 | 16 | 256k |
| Lunar Lander | 250 | 32 | 8M |
| Ant | 1k | 32 | 4M |

### B.1  Half Cheetah

The v4 version is used out-of-the-box.

### B.2  Cliff Walking

The v0 version is modified to have a fixed horizon of 250 timesteps and a custom reward function. The standard version has a reward of -1 every timestep with the episode terminating when the end is reached. Walking off the cliff gives -100 reward and returns the agent to the start. Our fixed horizon version of this is the same except reaching the end state does not terminate the environment, and instead grants 5 reward per timestep spent there. This was based on what lead to good learning with PPO and access to the reward function directly.

As the reward function is sparse, for sampling preferences only, a shaped version of it is used to simulate human intuition on what behaviours are closer to optimal. The penalty for walking off cliffs remains the same, but otherwise the agent receives a weighted reward of -1 and 5 depending on how close in $L_1$ norm it is to the start/end state respectively.

### B.3 Lunar Lander

515 The v2 version is modified to have a fixed horizon of 250 timesteps and a custom reward function.

516 The reward function used is mostly the same as in the Gymnasium version, except instead of
517 terminating on game over or the lander not being awake (i.e. landed), a -1 or +1 reward is issued each
518 timestep respectively.

### B.4 Ant

520 V4 version with `terminate_when_unhealthy=False` so that there are more maximum length
521 trajectories.

## C Supplementary Results

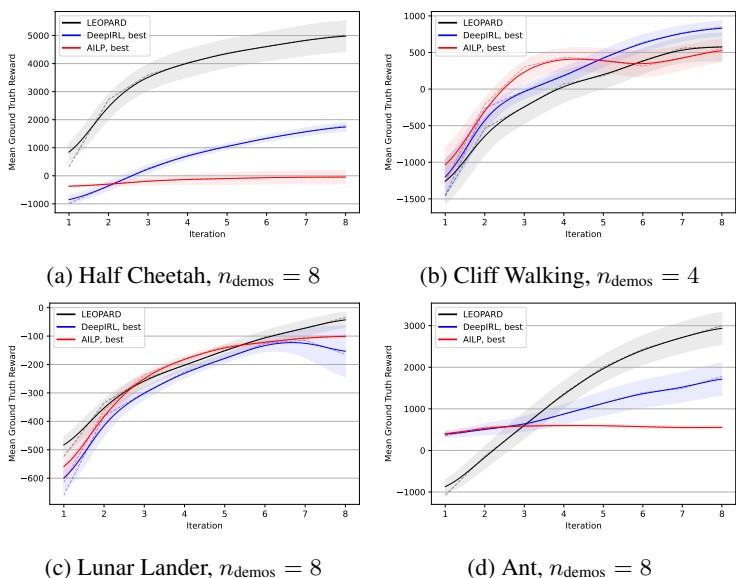

(a) Half Cheetah, $n_{\text{demos}} = 8$      (b) Cliff Walking, $n_{\text{demos}} = 4$

(c) Lunar Lander, $n_{\text{demos}} = 8$      (d) Ant, $n_{\text{demos}} = 8$

Figure 6: Comparison of LEOPARD with the baselines of AILP and DeepIRL when only positive demonstrations are available. The lines denote the mean of the ground truth reward function, with shaded standard errors across 16 random seeds, against algorithm iterations—alternations between optimising the reward model and the agent. Solid lines are smoothed means for clarity, dashed lines give raw values. A breakdown of the performance of the baseline methods for different reward model training epochs per iteration is given in Figures 9 and 10.

Table 2: Final ground truth reward to 3 s.f. with standard error for LEOPARD against a variety of baselines. (Top) 50/50 mix of preferences and positive demonstrations with baselines of AILP, performing DeepIRL followed by RLHF, and performing RLHF followed by DeepIRL (Half Cheetah only). See Figure 2 for reward vs algorithm iteration. (Bottom) Only positive demonstrations with baselines of AILP and DeepIRL. See Figure 6 for reward vs algorithm iteration. 'RM epochs per iter' is the number of training epochs for the reward model on each iteration of the algorithm, required to be fixed for DeepIRL. **Best** in column for section.

| Method | RM epochs per iter | Final Ground Truth Reward ± std error | | | |
|---|---|---|---|---|---|
| | | Half Cheetah | Cliff Walking | Lunar Lander | Ant |
| LEOPARD (ours) | Dynamic | **5650** ± 386 | **670** ± 116 | **-140** ± 49.8 | **2630** ± 322 |
| AILP | Dynamic | 3.49 ± 105 | -249 ± 6.09 | -684 ± 31.8 | -1130 ± 142 |
| AILP | 1 | 14.1 ± 234 | -266 ± 116 | -2010 ± 506 | -237 ± 110 |
| AILP | 2 | 25.1 ± 226 | -172 ± 74.2 | -2270 ± 507 | -300 ± 117 |
| AILP | 4 | -129 ± 35.9 | -181 ± 85.5 | -1930 ± 501 | 150 ± 131 |
| AILP | 8 | -87.0 ± 38.4 | -180 ± 70.0 | -813 ± 340 | 148 ± 55.0 |
| DeepIRL then RLHF | 1 | -389 ± 223 | -46.8 ± 125 | -2340 ± 548 | -766 ± 216 |
| DeepIRL then RLHF | 2 | 189 ± 312 | 1.34 ± 163 | -2200 ± 537 | -803 ± 259 |
| DeepIRL then RLHF | 4 | 224 ± 205 | -61.7 ± 115 | -2000 ± 467 | -792 ± 221 |
| DeepIRL then RLHF | 8 | 1540 ± 374 | -91.7 ± 103 | -1720 ± 548 | -927 ± 192 |
| LEOPARD (ours) | Dynamic | **5020** ± 555 | 580 ± 199 | **-34.4** ± 25.7 | **3000** ± 390 |
| AILP | Dynamic | -45.0 ± 236 | 554 ± 146 | -215 ± 16.1 | -489 ± 178 |
| AILP | 1 | -88.3 ± 9.15 | 381 ± 131 | -99.5 ± 5.45 | 555 ± 37.1 |
| AILP | 2 | -61.5 ± 47.1 | 330 ± 156 | -131 ± 9.33 | 450 ± 54.8 |
| AILP | 4 | -118 ± 6.08 | 205 ± 133 | -180 ± 12.3 | 300 ± 79.1 |
| AILP | 8 | -96.2 ± 6.36 | -72.2 ± 93.2 | -214 ± 8.62 | 268 ± 59.4 |
| DeepIRL | 1 | 1470 ± 318 | 828 ± 92.2 | -575 ± 194 | -295 ± 230 |
| DeepIRL | 2 | 1610 ± 264 | 769 ± 111 | -164 ± 98.6 | 1320 ± 426 |
| DeepIRL | 4 | 1290 ± 216 | **849** ± 102 | -159 ± 18.0 | 1780 ± 399 |
| DeepIRL | 8 | 1790 ± 162 | 528 ± 105 | -219 ± 21.3 | 1340 ± 319 |

Table 3: Final ground truth reward with standard error for LEOPARD across a variety of mixture of types of feedback. For details on feedback amounts per environment and the reward vs algorithm iteration see Figure 3. **Best** in column.

| Feedback types | Final Ground Truth Reward ± std error | | | |
|---|---|---|---|---|
| | Half Cheetah | Cliff Walking | Lunar Lander | Ant |
| Preferences | 4960 ± 574 | -252 ± 2.22 | -163 ± 19.7 | 1510 ± 491 |
| Positive demonstrations | 5020 ± 555 | 580 ± 199 | **-34.4** ± 25.7 | **3000** ± 390 |
| Preferences and positive demos | **5650** ± 386 | 670 ± 116 | -140 ± 49.8 | 2630 ± 322 |
| Positive and negative demos | 2870 ± 609 | **883** ± 79.0 | -169 ± 107 | 754 ± 339 |
| Prefs, pos and neg demos | 3640 ± 603 | 514 ± 133 | -120 ± 11.3 | 1580 ± 296 |

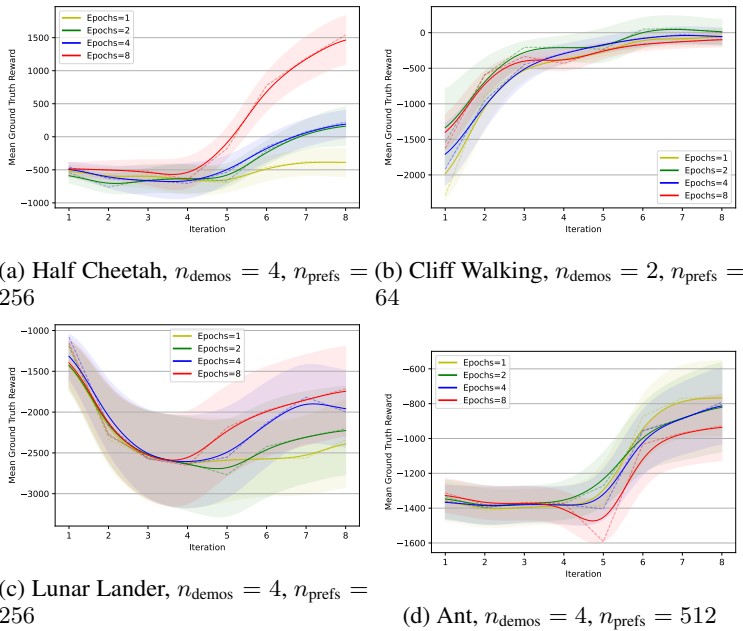

(a) Half Cheetah, $n_{\text{demos}} = 4$, $n_{\text{prefs}} = 256$

(b) Cliff Walking, $n_{\text{demos}} = 2$, $n_{\text{prefs}} = 64$

(c) Lunar Lander, $n_{\text{demos}} = 4$, $n_{\text{prefs}} = 256$

(d) Ant, $n_{\text{demos}} = 4$, $n_{\text{prefs}} = 512$

Figure 7: Breakdown of the DeepIRL followed by RLHF baseline, for different numbers of epochs that the reward model was trained for per algorithm iteration. The lines denote the mean of the ground truth reward function, with shaded standard errors across 16 random seeds, against algorithm iterations. Solid lines are smoothed means for clarity, dashed lines give raw values.

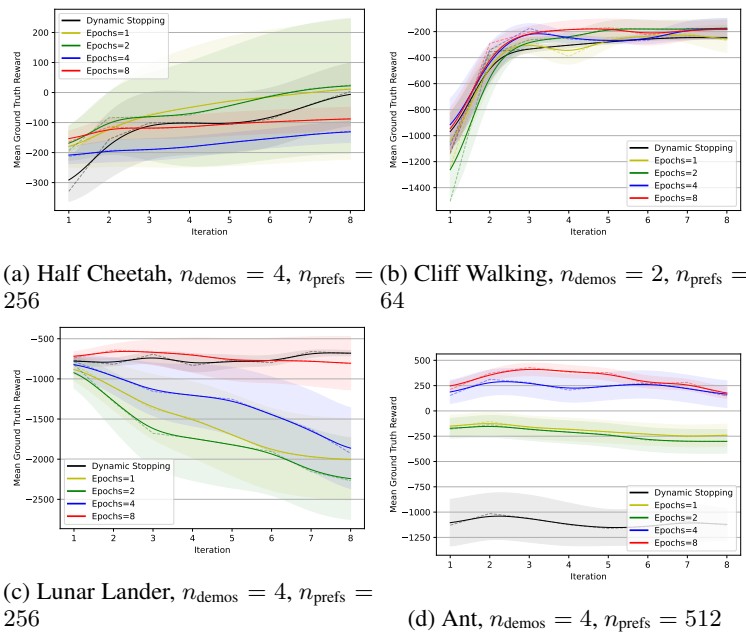

(a) Half Cheetah, $n_{\text{demos}} = 4$, $n_{\text{prefs}} = 256$

(b) Cliff Walking, $n_{\text{demos}} = 2$, $n_{\text{prefs}} = 64$

(c) Lunar Lander, $n_{\text{demos}} = 4$, $n_{\text{prefs}} = 256$

(d) Ant, $n_{\text{demos}} = 4$, $n_{\text{prefs}} = 512$

Figure 8: Breakdown of the AILP baseline for positive demonstrations and preferences, for different numbers of epochs that the reward model was trained for per algorithm iteration. The lines denote the mean of the ground truth reward function, with shaded standard errors across 16 random seeds, against algorithm iterations. Solid lines are smoothed means for clarity, dashed lines give raw values.

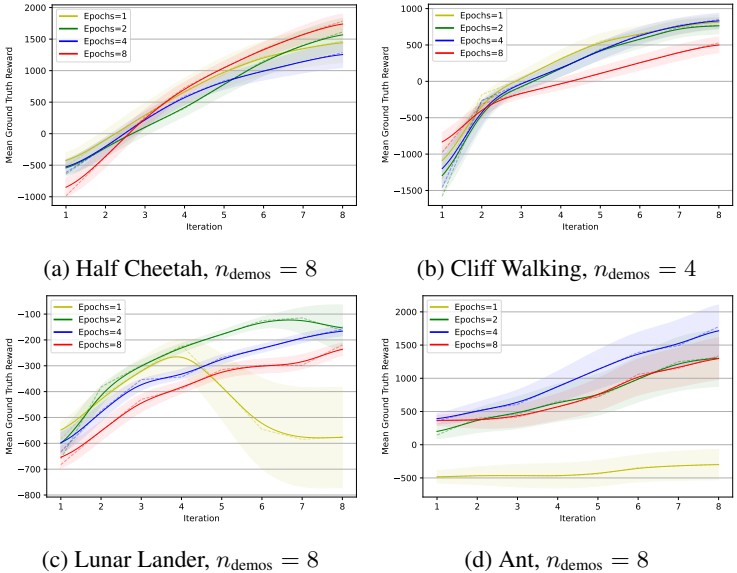

(a) Half Cheetah, $n_{\text{demos}} = 8$      (b) Cliff Walking, $n_{\text{demos}} = 4$

(c) Lunar Lander, $n_{\text{demos}} = 8$      (d) Ant, $n_{\text{demos}} = 8$

Figure 9: Breakdown of the DeepIRL baseline, for different numbers of epochs that the reward model was trained for per algorithm iteration. The lines denote the mean of the ground truth reward function, with shaded standard errors across 16 random seeds, against algorithm iterations. Solid lines are smoothed means for clarity, dashed lines give raw values.

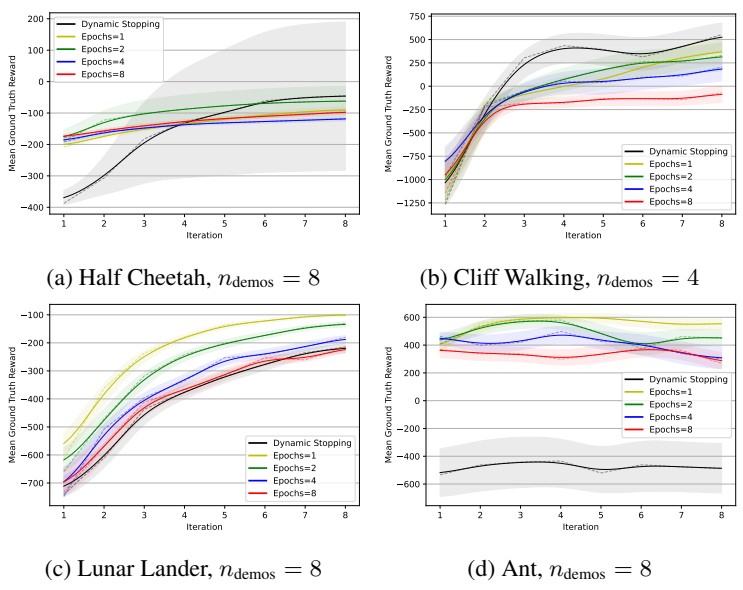

(a) Half Cheetah, $n_{\text{demos}} = 8$      (b) Cliff Walking, $n_{\text{demos}} = 4$

(c) Lunar Lander, $n_{\text{demos}} = 8$      (d) Ant, $n_{\text{demos}} = 8$

Figure 10: Breakdown of the AILP baseline for positive demonstrations only, for different numbers of epochs that the reward model was trained for per algorithm iteration. The lines denote the mean of the ground truth reward function, with shaded standard errors across 16 random seeds, against algorithm iterations. Solid lines are smoothed means for clarity, dashed lines give raw values.

Table 4: Outliers for Cliff Walking that were removed from the main analysis. This is defined as having less than -3000 reward on any iteration from the second onwards. Note there were 16 random seeds in total. If multiple 'RM epochs per iteration's are given, this is the total across them all.

| Method | RM epochs per iteration | Cliff Walking Outliers (%) |
|---|---|---|
| LEOPARD (preferences only) | Dynamic | 0 (0%) |
| LEOPARD (positive demonstrations only) | Dynamic | 3 (19%) |
| LEOPARD (positive demonstrations and preferences) | Dynamic | 2 (13%) |
| LEOPARD (mixed demonstrations) | Dynamic | 0 (0%) |
| LEOPARD (mixed demonstrations and preferences) | Dynamic | 2 (13%) |
| AILP (positive demonstrations only) | Dynamic, 1, 2, 4, 8 | 0 (0%) |
| AILP (positive demonstrations and preferences) | Dynamic | 1 (6%) |
| AILP (positive demonstrations and preferences) | 1, 2, 4, 8 | 0 (0%) |
| DeepIRL only | 1, 2, 4, 8 | 0 (0%) |
| DeepIRL then RLHF | 1 | 3 (19%) |
| DeepIRL then RLHF | 2 | 7 (44%) |
| DeepIRL then RLHF | 4 | 4 (25%) |
| DeepIRL then RLHF | 8 | 1 (6%) |

# D   Main Proofs

Here we more stringently define and prove the theoretical result from the end of Section 3.1, and then prove the models considered in Appendix E do not satisfy it.

**Theorem D.1.** *Upper bounds on RRPO loss give lower bounds on reward difference of related fragments. For all $\epsilon > 0$, if $\mathcal{L}_{RRPO} \leq \epsilon$, then for all $\tau_a, \tau_b \in \mathcal{D}^2$ where there exists a $<_x \in \mathcal{C}$ such that $\tau_a <_x \tau_b$, we have the following:*

$$R_\theta(\tau_b) - R_\theta(\tau_a) > -\frac{1}{\beta_x} \log(e^\epsilon - 1), \tag{11}$$

*where $\beta_x$ is the rationality coefficient of $<_x$.*

*Proof.* We will prove this by contrapositive, that is if:

$$R_\theta(\tau_b) - R_\theta(\tau_a) \leq -\frac{1}{\beta_x} \log(e^\epsilon - 1), \tag{12}$$

for some $\epsilon > 0$, and there exists a $<_x$ such that $\tau_a <_x \tau_b$, then $\mathcal{L}_{\text{RRPO}} > \epsilon$.

Assume Equation (12) and that the relevant $<_x$ exists. Consider Equation (5):

$$\mathcal{L}_{\text{RRPO}}(\theta) = -\log P_{\text{RRPO}}(\mathcal{C}|\mathcal{D}, \theta)$$

$$= -\sum_{(\tau_i, <_j) \in \mathcal{D} \times \mathcal{C}} \log \frac{\exp(\beta_j R_\theta(\tau_i))}{\exp(\beta_j R_\theta(\tau_i)) + \sum_{\tau_k \in \mathcal{D}} \mathbf{1}_{\tau_k <_j \tau_i} \exp(\beta_j R_\theta(\tau_k))}$$

$$= \sum_{(\tau_i, <_j) \in \mathcal{D} \times \mathcal{C}} \log \frac{\exp(\beta_j R_\theta(\tau_i)) + \sum_{\tau_k \in \mathcal{D}} \mathbf{1}_{\tau_k <_j \tau_i} \exp(\beta_j R_\theta(\tau_k))}{\exp(\beta_j R_\theta(\tau_i))}$$

$$= \sum_{(\tau_i, <_j) \in \mathcal{D} \times \mathcal{C}} \log \left( 1 + \frac{\sum_{\tau_k \in \mathcal{D}} \mathbf{1}_{\tau_k <_j \tau_i} \exp(\beta_j R_\theta(\tau_k))}{\exp(\beta_j R_\theta(\tau_i))} \right).$$

Consider the term $(\tau_b, <_x)$, and bring it outside the summation.

$$\mathcal{L}_{\text{RRPO}}(\theta) = \log \left( 1 + \frac{\sum_{\tau_k \in \mathcal{D}} \mathbf{1}_{\tau_k <_x \tau_b} \exp(\beta_x R_\theta(\tau_k))}{\exp(\beta_x R_\theta(\tau_b))} \right) + \sum_{\substack{(\tau_i, <_j) \in \mathcal{D} \times \mathcal{C} \\ (\tau_i, <_j) \neq (\tau_b, <_x)}} \log(1 + ...).$$

The remaining terms are strictly positive, and $\mathbf{1}_{\tau_a <_x \tau_b} = 1$.

$$\mathcal{L}_{\text{RRPO}}(\theta) > \log \left( 1 + \frac{\exp(\beta_x R_\theta(\tau_a)) + ...}{\exp(\beta_x R_\theta(\tau_b))} \right)$$

$$= \log \left( 1 + \exp(\beta_x R_\theta(\tau_a) - \beta_x R_\theta(\tau_b)) + \frac{...}{\exp(\beta_x R_\theta(\tau_b))} \right)$$

$$> \log \left( 1 + \exp(\beta_x (R_\theta(\tau_a) - R_\theta(\tau_b))) \right),$$

by ignoring terms that are strictly positive. Sub in Equation (12).

$$\mathcal{L}_{\text{RRPO}}(\theta) > \log \left( 1 + \exp \left( \beta_x \left( \frac{1}{\beta_x} \log(e^\epsilon - 1) \right) \right) \right)$$

$$= \log(1 + e^\epsilon - 1)$$

$$= \epsilon,$$

as required. □

Consider a special case where $\epsilon = \log 2$, Equation (11) becomes:

$$R_\theta(\tau_b) - R_\theta(\tau_a) > -\frac{1}{\beta_x} \log(e^{\log 2} - 1)$$

$$= 0,$$

$$\therefore R_\theta(\tau_b) > R_\theta(\tau_a).$$

## E  Alternative RRC-Derived Approaches

RRPO and LEOPARD are very simple and natural extensions of existing work, however, they are not trivially so. Building off RRC, there are several approaches to preference and demonstration learning that appear natural and are simple, and yet are deficient. Here we explore two of them in the preference and ranked positive demonstrations only setting.

Let the notation be as defined in Section 3.2. We will assume that preferences, positive demonstration selection, and the rankings over the positive demonstrations are all independent. Our overall likelihood function shall be:

$$
\begin{aligned}
P_{\text{Feedback}}(\mathcal{C}|\mathcal{D}, \theta) = P_{\text{Pos-Demo}}(\mathcal{D}_{\text{pos}} \succ \mathcal{D}_{\text{agent}}|\mathcal{D}_{\text{pos}}, \mathcal{D}_{\text{agent}}, \theta) \\
\cdot P_{\text{Rank}}(<_{\text{pos}}|\mathcal{D}_{\text{pos}}, \theta) \\
\cdot \prod_{(\tau_a, \tau_b) \in \mathcal{P}} P_{\text{RLHF}}(\tau_a \succ \tau_b|\theta),
\end{aligned} \tag{13}
$$

where $P_{\text{Rank}}$ is something sensible.

We consider two potential candidates for $P_{\text{Pos-Demo}}$ derived via RRC in a simple manner:

$$
P_{\text{Sum-of-Choices}}(...) = \sum_{\tau \in \mathcal{D}_{\text{pos}}} P_{\text{RRC}}(C_{\tau}|\mathcal{D}_{\text{pos}} \cup \mathcal{D}_{\text{agent}}, \theta), \tag{14}
$$

$$
P_{\text{Choose-Best-Average}}(...) = P_{\text{RRC}}(C_{\text{Avg}(\mathcal{D}_{\text{pos}})}|\{\text{Avg}(\mathcal{D}_{\text{pos}}), \text{Avg}(\mathcal{D}_{\text{agent}})\}, \theta). \tag{15}
$$

Thus:

$$
P_{\text{Sum-of-Choices}}(...) = \frac{\sum_{\tau \in \mathcal{D}_{\text{pos}}} \exp(R_{\theta}(\tau))}{\sum_{\tau \in \mathcal{D}_{\text{pos}}} \exp(R_{\theta}(\tau)) + \sum_{\tau \in \mathcal{D}_{\text{agent}}} \exp(R_{\theta}(\tau))}, \tag{16}
$$

$$
P_{\text{Choose-Best-Average}}(...) = \frac{\exp\left(\frac{1}{|\mathcal{D}_{\text{pos}}|} \sum_{\tau \in \mathcal{D}_{\text{pos}}} R_{\theta}(\tau)\right)}{\exp\left(\frac{1}{|\mathcal{D}_{\text{pos}}|} \sum_{\tau \in \mathcal{D}_{\text{pos}}} R_{\theta}(\tau)\right) + \exp\left(\frac{1}{|\mathcal{D}_{\text{agent}}|} \sum_{\tau \in \mathcal{D}_{\text{agent}}} R_{\theta}(\tau)\right)}, \tag{17}
$$

with

$$
\mathcal{L}_{\text{SoC}} = -\log P_{\text{Sum-of-Choices}}, \tag{18}
$$

$$
\mathcal{L}_{\text{CBA}} = -\log P_{\text{Choose-Best-Average}}. \tag{19}
$$

Rationality coefficients are omitted since they are not critical to this analysis. We shall show that these models have undesirable theoretical properties, and poorer empirical performance compared to LEOPARD.

### E.1  Theoretical Properties

Neither $P_{\text{Sum-of-Choices}}$ nor $P_{\text{Choose-Best-Average}}$ have the property that upper bounds on their negative-log-likelihood give rise to lower bounds on reward differences between demonstrated trajectories and ones sampled from the agent, unlike $P_{\text{RRPO}}$. We prove this in Theorems E.1 and E.2 in Section E.2.1. Whilst this may not seem too critical, its combination with the potential effects of $P_{\text{Rank}}$, and its interaction with exploration in RL, can cause a very undesirable failure mode.

Imagine an environment where three distinct behaviours are possible, A, B, and C. We prefer C to B, and B to A, so we provide a demonstration of B and C each, $\tau_b, \tau_c$, and express via the ranking model that $\tau_c \succ \tau_b$. This ranking is fitted by assigning high reward to C, and low to B. Our agent is initialised generating from A. Our demonstration model, seeing $\tau_c$ have high reward, does not lower the reward of A that much, and does not mind that $\tau_b$ has low reward. We're left with low loss and yet a reward model that could prefer A to B.

Now consider that our environment has some unfavourable dynamics. Policies that generate A, are quite different from those that generate C, with B being somewhere between the two. Thus, to eventually generate C, our policy will first need to explore B. However, our reward model gives it lower reward when it tries this, and so the agent sticks to what it thinks is best, behaviour A, much to our disappointment.

570 Whilst a little contrived, the above story highlights a certain failure mode that could occur if one
571 combined demonstration rankings with a demonstration model that does not satisfy Theorem D.1.
572 If it did satisfy it, such as for RRPO and LEOPARD, then low loss cannot be achieved unless the
573 reward model prefers B to A, preventing the issue.

574 Alleviating this problem by omitting the rankings is suboptimal, as we lose information. However,
575 $P_{\text{Sum-of-Choices}}$ suffers further. It is shown in Section E.2.2 that the gradient of $\mathcal{L}_{\text{SoC}}$ with respect to $\theta$
576 can be expressed in the following form.

$$-\frac{\partial}{\partial\theta}\mathcal{L}_{\text{SoC}} = \sum_{\tau_a \in \mathcal{D}_{\text{agent}}} P_{\text{RRC}}(C_a|\mathcal{T}, \theta) \left( \sum_{\tau_p \in \mathcal{D}_{\text{pos}}} P_{\text{RRC}}(C_p|\mathcal{D}_{\text{pos}}, \theta) \frac{\partial}{\partial\theta} R_\theta(\tau_p) - \frac{\partial}{\partial\theta} R_\theta(\tau_a) \right), \quad (20)$$

577 where $C_i$ is the human choice for $\tau_i$, and $\mathcal{T} = \mathcal{D}_{\text{pos}} \cup \mathcal{D}_{\text{agent}}$. We see that the reward of agent
578 trajectories are pushed down proportional to the probability that they would be chosen out of the
579 combined set of trajectories. This makes sense—if our reward model thinks highly of specific agent
580 trajectories, it ought to adjust its beliefs so that it no longer favours them.

581 However, the demonstration trajectories are also pushed up in reward proportional to the probability
582 that they would be chosen. That is to say, the better the reward model thinks the demonstrated
583 trajectory is, the more it thinks it should increase its reward, a positive feedback loop! In practice,
584 the reward model is going to have some initial preferences over the demonstrated trajectories due to
585 its initialisation. Since this will be random, it will most likely be incorrect. It will then proceed to
586 reinforce its own incorrect beliefs and lock-in its own ranking of the demonstrations. This means
587 our reward model will not provide correct rewards to guide the agent towards better behaviour in
588 the trajectory space around the demonstrations. Furthermore, if it generalises from these incorrect
589 beliefs, it could also become wrong about other parts of trajectory space, further reducing the quality
590 of the reward signal for the agent.

## E.2 Chapter Proofs and Derivations

### E.2.1 Reward Bounds

593 **Theorem E.1.** *Upper bounds on Sum-of-Choices loss do not give lower bounds on reward difference*
594 *between demonstrations and agent trajectories. For all $\epsilon > 0$, if $\mathcal{L}_{SoC} \leq \epsilon$, we cannot guarantee that*

$$R_\theta(\tau_p) - R_\theta(\tau_a) > f(\epsilon) \quad (21)$$

595 *for all $\tau_p, \tau_a \in \mathcal{D}_{pos} \times \mathcal{D}_{agent}$, where $f$ is a function of type $\mathbb{R}^+ \to \mathbb{R}$.*

596 *Proof.* We will prove this by example.

597 Consider

$$\begin{aligned}
\mathcal{D}_{\text{pos}} &= \{\tau_1, \tau_2\}, \\
\mathcal{D}_{\text{agent}} &= \{\tau_a\}, \\
R_\theta(\tau_1) &= r_1, \\
R_\theta(\tau_2) &= r_2, \\
R_\theta(\tau_a) &= r_a.
\end{aligned}$$

598 We now expand Equation (18) with Equation (16) and the above.

$$\begin{aligned}
\mathcal{L}_{\text{SoC}}(\theta) &= -\log\left(\frac{e^{r_1} + e^{r_2}}{e^{r_1} + e^{r_2} + e^{r_a}}\right) \\
&= \log\left(1 + \frac{e^{r_a}}{e^{r_1} + e^{r_2}}\right).
\end{aligned}$$

599 Assume $\mathcal{L}_{\text{SoC}} \leq \epsilon$, therefore

$$\begin{aligned}
\log\left(1 + \frac{e^{r_a}}{e^{r_1} + e^{r_2}}\right) &\leq \epsilon, \\
r_a &\leq \log\left((e^\epsilon - 1)(e^{r_1} + e^{r_2})\right).
\end{aligned}$$

Let

$$r_a = \log\left((e^\epsilon - 1)(e^{r_1} + e^{r_2})\right).$$

Consider $r_1 - r_a$, substituting in the above expression:

$$\begin{aligned}
r_1 - r_a &= r_1 - \log((e^\epsilon - 1)(e^{r_1} + e^{r_2})) \\
&= r_1 - \log(e^\epsilon - 1) - \log(e^{r_1} + e^{r_2}) \\
&\leq r_1 - \log(e^\epsilon - 1) - r_2,
\end{aligned}$$

as $\log(x + y) \geq \log(y)$ for positive $x$ and $y$. Thus, we see that for a fixed $r_1$ and $\epsilon$, we can choose $r_2$ and $r_a$ such that $\mathcal{L}_{\text{SoC}} \leq \epsilon$, but $r_1 - r_a$ can be arbitrarily negative. $\square$

**Theorem E.2.** *Upper bounds on Choose-Best-Average loss do not give lower bounds on reward difference between demonstrations and agent trajectories. For all $\epsilon > 0$, if $\mathcal{L}_{CBA} \leq \epsilon$, we cannot guarantee that*

$$R_\theta(\tau_p) - R_\theta(\tau_a) > f(\epsilon) \tag{22}$$

*for all $\tau_p, \tau_a \in \mathcal{D}_{pos} \times \mathcal{D}_{agent}$, where $f$ is a function of type $\mathbb{R}^+ \to \mathbb{R}$.*

*Proof.* We will proceed similarly to the above, assuming the same notation.

Expanding Equation (19) with Equation (17).

$$\begin{aligned}
\mathcal{L}_{\text{CBA}}(\theta) &= -\log\left(\frac{\exp\left(\frac{1}{2}(r_1 + r_2)\right)}{\exp\left(\frac{1}{2}(r_1 + r_2)\right) + \exp(r_a)}\right) \\
&= \log\left(1 + \frac{\exp(r_a)}{\exp\left(\frac{1}{2}(r_1 + r_2)\right)}\right) \\
&= \log\left(1 + \exp\left(r_a - \frac{1}{2}(r_1 + r_2)\right)\right).
\end{aligned}$$

Assume $\mathcal{L}_{\text{CBA}} \leq \epsilon$, therefore

$$\log\left(1 + \exp\left(r_a - \frac{1}{2}(r_1 + r_2)\right)\right) \leq \epsilon,$$

$$r_a \leq \log(e^\epsilon - 1) + \frac{1}{2}(r_1 + r_2).$$

Let

$$r_a = \log(e^\epsilon - 1) + \frac{1}{2}(r_1 + r_2).$$

Consider $r_1 - r_a$, substituting in the above expression:

$$r_1 - r_a = r_1 - \log(e^\epsilon - 1) - \frac{1}{2}(r_1 + r_2).$$

Again, we see that for a fixed $r_1$ and $\epsilon$, we can choose $r_2$ and $r_a$ such that $\mathcal{L}_{\text{SoC}} \leq \epsilon$, but $r_1 - r_a$ can be arbitrarily negative. $\square$

### E.2.2 Loss Gradients

Here we will show that the gradient with respect to $\theta$ of $\mathcal{L}_{\text{SoC}}$ can be expressed in the form given in Equation (20) of Section E.1.

First we give a simplification of deterministic RRC with $\beta = 1$ and $\psi(x) = x$ for all $x$, and some additional notation:

$$C : () \to \mathcal{D},$$

$$P_{\text{RRC}}(C_i|\mathcal{D}, \theta) = \frac{e^{R_\theta(\tau_i)}}{\sum_{\tau_j \in \mathcal{D}} e^{R_\theta(\tau_j)}},$$

$$\mathcal{T} = \mathcal{D}_{\text{pos}} \cup \mathcal{D}_{\text{agent}}.$$

Now we derive some useful identities.

$$\frac{\partial}{\partial \theta} \log \sum_{\tau \in \mathcal{D}} e^{R_\theta(\tau)} = \frac{\frac{\partial}{\partial \theta} \sum_{\tau_i \in \mathcal{D}} e^{R_\theta(\tau_i)}}{\sum_{\tau_j \in \mathcal{D}} e^{R_\theta(\tau_j)}}$$

$$= \sum_{\tau_i \in \mathcal{D}} \frac{\frac{\partial}{\partial \theta} e^{R_\theta(\tau_i)}}{\sum_{\tau_j \in \mathcal{D}} e^{R_\theta(\tau_j)}}$$

$$= \sum_{\tau_i \in \mathcal{D}} \frac{e^{R_\theta(\tau_i)}}{\sum_{\tau_j \in \mathcal{D}} e^{R_\theta(\tau_j)}} \frac{\partial}{\partial \theta} R_\theta(\tau_i)$$

$$= \sum_{\tau_i \in \mathcal{D}} P_{\text{RRC}}(C_i | \mathcal{D}, \theta) \frac{\partial}{\partial \theta} R_\theta(\tau_i), \tag{23}$$

$$P_{\text{RRC}}(C_i | \mathcal{A}, \theta) = \frac{e^{R_\theta(\tau_i)}}{\sum_{\tau_j \in \mathcal{A}} e^{R_\theta(\tau_j)}}$$

$$= \frac{e^{R_\theta(\tau_i)}}{\sum_{\tau_j \in \mathcal{A}} e^{R_\theta(\tau_j)}} \frac{\sum_{\tau_k \in \mathcal{A} \cup \mathcal{B}} e^{R_\theta(\tau_k)}}{\sum_{\tau_k \in \mathcal{A} \cup \mathcal{B}} e^{R_\theta(\tau_k)}}$$

$$= \frac{P_{\text{RRC}}(C_i | \mathcal{A} \cup \mathcal{B}, \theta)}{\sum_{\tau_j \in \mathcal{A}} P_{\text{RRC}}(C_j | \mathcal{A} \cup \mathcal{B}, \theta)}, \tag{24}$$

$$P_{\text{RRC}}(C_i | \mathcal{A}, \theta) - P_{\text{RRC}}(C_i | \mathcal{A} \cup \mathcal{B}, \theta) = \frac{P_{\text{RRC}}(C_i | \mathcal{A} \cup \mathcal{B}, \theta)}{\sum_{\tau_j \in \mathcal{A}} P_{\text{RRC}}(C_j | \mathcal{A} \cup \mathcal{B}, \theta)} - P_{\text{RRC}}(C_i | \mathcal{A} \cup \mathcal{B}, \theta)$$

$$= \frac{P_{\text{RRC}}(C_i | \mathcal{A} \cup \mathcal{B}, \theta) \left(1 - \sum_{\tau_j \in \mathcal{A}} P_{\text{RRC}}(C_i | \mathcal{A} \cup \mathcal{B}, \theta)\right)}{\sum_{\tau_j \in \mathcal{A}} P_{\text{RRC}}(C_j | \mathcal{A} \cup \mathcal{B}, \theta)}$$

$$= \frac{P_{\text{RRC}}(C_i | \mathcal{A} \cup \mathcal{B}, \theta) \sum_{\tau_k \in \mathcal{B}} P_{\text{RRC}}(C_k | \mathcal{A} \cup \mathcal{B}, \theta)}{\sum_{\tau_j \in \mathcal{A}} P_{\text{RRC}}(C_j | \mathcal{A} \cup \mathcal{B}, \theta)}$$

$$= \sum_{\tau_k \in \mathcal{B}} P_{\text{RRC}}(C_k | \mathcal{A} \cup \mathcal{B}, \theta) \frac{P_{\text{RRC}}(C_i | \mathcal{A} \cup \mathcal{B}, \theta)}{\sum_{\tau_j \in \mathcal{A}} P_{\text{RRC}}(C_j | \mathcal{A} \cup \mathcal{B}, \theta)}$$

$$= \sum_{\tau_k \in \mathcal{B}} P_{\text{RRC}}(C_k | \mathcal{A} \cup \mathcal{B}, \theta) P_{\text{RRC}}(C_i | \mathcal{A}, \theta) \tag{25}$$

Now we use these identities to derive the special form of the gradient of $\mathcal{L}_{\text{SoC}}$.

$$
\begin{aligned}
-\frac{\partial}{\partial\theta}\mathcal{L}_{\text{SoC}} =& \frac{\partial}{\partial\theta}\log\frac{\sum_{\tau\in\mathcal{D}_{\text{pos}}}e^{R_\theta(\tau)}}{\sum_{\tau\in\mathcal{D}_{\text{pos}}}e^{R_\theta(\tau)}+\sum_{\tau\in\mathcal{D}_{\text{agent}}}e^{R_\theta(\tau)}} \\
=& \frac{\partial}{\partial\theta}\log\sum_{\tau\in\mathcal{D}_{\text{pos}}}e^{R_\theta(\tau)}-\frac{\partial}{\partial\theta}\log\sum_{\tau\in\mathcal{T}}e^{R_\theta(\tau)} \\
=& \sum_{\tau_p\in\mathcal{D}_{\text{pos}}}P_{\text{RRC}}(C_p|\mathcal{D}_{\text{pos}},\theta)\frac{\partial}{\partial\theta}R_\theta(\tau_p)-\sum_{\tau_i\in\mathcal{T}}P_{\text{RRC}}(C_i|\mathcal{T},\theta)\frac{\partial}{\partial\theta}R_\theta(\tau_i) \\
=& \sum_{\tau_p\in\mathcal{D}_{\text{pos}}}P_{\text{RRC}}(C_p|\mathcal{D}_{\text{pos}},\theta)\frac{\partial}{\partial\theta}R_\theta(\tau_p)-\sum_{\tau_p\in\mathcal{D}_{\text{pos}}}P_{\text{RRC}}(C_p|\mathcal{T},\theta)\frac{\partial}{\partial\theta}R_\theta(\tau_p) \\
& -\sum_{\tau_a\in\mathcal{D}_{\text{agent}}}P_{\text{RRC}}(C_a|\mathcal{T},\theta)\frac{\partial}{\partial\theta}R_\theta(\tau_a) \\
=& \sum_{\tau_p\in\mathcal{D}_{\text{pos}}}\left(P_{\text{RRC}}(C_p|\mathcal{D}_{\text{pos}},\theta)-P_{\text{RRC}}(C_p|\mathcal{T},\theta)\right)\frac{\partial}{\partial\theta}R_\theta(\tau_p) \\
& -\sum_{\tau_a\in\mathcal{D}_{\text{agent}}}P_{\text{RRC}}(C_a|\mathcal{T},\theta)\frac{\partial}{\partial\theta}R_\theta(\tau_a) \\
=& \sum_{\tau_p\in\mathcal{D}_{\text{pos}}}\sum_{\tau_a\in\mathcal{D}_{\text{agent}}}P_{\text{RRC}}(C_a|\mathcal{T},\theta)P_{\text{RRC}}(C_p|\mathcal{D}_{\text{pos}},\theta)\frac{\partial}{\partial\theta}R_\theta(\tau_p) \\
& -\sum_{\tau_a\in\mathcal{D}_{\text{agent}}}P_{\text{RRC}}(C_a|\mathcal{T},\theta)\frac{\partial}{\partial\theta}R_\theta(\tau_a) \\
=& \sum_{\tau_a\in\mathcal{D}_{\text{agent}}}P_{\text{RRC}}(C_a|\mathcal{T},\theta)\left(\sum_{\tau_p\in\mathcal{D}_{\text{pos}}}P_{\text{RRC}}(C_p|\mathcal{D}_{\text{pos}},\theta)\frac{\partial}{\partial\theta}R_\theta(\tau_p)-\frac{\partial}{\partial\theta}R_\theta(\tau_a)\right). \quad (26)
\end{aligned}
$$

## F  Impact Statement

This paper aims to improve our ability to leverage diverse ranges of feedback when training reward models for RL agents with difficult to specify objectives. This will hopefully lead to reward models that are more accurate and robust, reducing problems such as specification gaming and sensitivity to noise. In turn, this will make deployed systems that have been trained in this manner more aligned and less likely fail.

## G  Use of Existing Assets

The primary assets we use in this work are referenced in Table 5 alongside their respective licenses.

Table 5: Assets used in the paper.

| Name | Type | Reference | License |
|------|------|-----------|---------|
| Stable Baselines3 | Code | Raffin et al. (2021) | MIT |
| RL Baselines3 Zoo | Hyperparameters | Raffin (2020) | MIT |
| Gymnasium | Code | Towers et al. (2024) | MIT |
| MuJoCo | Code | Todorov et al. (2012) | Apache-2.0 |
| JAX | Code | Bradbury et al. (2018) | Apache-2.0 |
| Flax | Code | Heek et al. (2024) | Apache-2.0 |
| Optax | Code | DeepMind et al. (2020) | Apache-2.0 |

# H   Use of Compute

We ran all of our experiments on a CPU server with 4 cores each. Approximate wallclock runtime for each environment was as follows: Cliffwalking 10 minutes; Half Cheetah 1 hour; Lunar Lander 1 hour 45 minutes; Ant 2 hours. This runtime was consistent regardless of the reward learning method or amount of feedback, as the agent-environment interaction time and reinforcement learning was where most time was spent each run. Thus the CPU-core hours per run for each environment were: Cliffwalking 40 minutes; Half Cheetah 4 hours; Lunar Lander 7 hours; Ant 8 hours. For each combination of algorithm, feedback mixture, and environment, we ran 16 seeds to get reliable results.

For the comparisons to baseline, we had to run two sets of seeds per environment to benchmark LEOPARD, and 18 sets of seeds per environment to benchmark the baselines.[13] With 16 seeds per combination this totals to the following CPU-core hours for each environment for this set of experiments: Cliffwalking 213 hours, Half Cheetah 1280 hours, Lunar Lander 2240 hours, Ant 2560 hours. The total for this experiment is thus 6293 CPU-core hours.

For the feedback mixture experiments we had to run five sets of seeds per environment to benchmark each of the five feedback type mixtures with LEOPARD. With 16 seeds per combination this totals to the following CPU-core hours for each environment for this set of experiments: Cliffwalking 53 hours, Half Cheetah 320 hours, Lunar Lander 560 hours, Ant 640 hours. The total for this experiment is thus 1573 CPU-core hours.

The total for all experiments is therefore 7866 CPU-core hours.

Further CPU hours were also used to obtain the synthetic demonstrations and for preliminary experiments. The synthetic demonstration generation time was negligible, and preliminary experiments added at most 2x to the total compute used, bringing it to around 24k CPU-core hours for the whole project.

---

[13]This is due to the fact that LEOPARD uses early stopping for reward model training whereas the baselines either couldn't, or could've likely performed better with a fixed number of training epochs. Thus a small number of reward model training epochs had to be swept over for each baseline.

