# OpenReview forum: "Learning from Preferences and Mixed Demonstrations in General Settings"
_NeurIPS.cc/2025/Conference — Submitted to NeurIPS 2025_

### Official Review · Reviewer_VZqx · 2025-07-02

**Clarity:** 2
**Significance:** 2
**Originality:** 2
**Rating:** 3
**Confidence:** 3

**Summary:**

The authors introduce a unifying framework called Reward-Rational Partial Orderings (RRPO), which models diverse feedback as Boltzmann-rational orderings over trajectory fragments. Building on RRPO, they propose LEOPARD, which jointly optimizes a reward model and agent policy by iteratively incorporating all available feedback types. In evaluations, LEOPARD consistently outperforms standard baselines that use demonstrations, then preferences, and other combined feedback methods, especially when feedback is scarce or mixed. Moreover, experiments show that leveraging a combination of feedback types—even limited negative demonstrations alongside preferences—yields more robust reward learning than using any single feedback source.

**Questions:**

- **Multi-Objective Trade-offs:** As tasks grow more complex, how does LEOPARD’s single loss function handle the inherent trade-offs of optimizing over multiple feedback types simultaneously?
- **Feedback Budget Selection:** What guided the choice of $n_{\mathrm{demos}}$ and $n_{\mathrm{prefs}}$ in your experiments? Are these settings transferable across domains?
- **Guaranteeing Balanced Mixing:** In Section 4.3, optimal data-mixing strategies differ by domain. How can one ensure that all feedback types are leveraged appropriately without manual tuning?
- **Baseline Comparison:** How does LEOPARD perform relative to the limited-preference imitation learning [1]?

**References**\
[1] Xingchen Cao et al., Limited Preference Aided Imitation Learning from Imperfect Demonstrations, ICML 2024.

**Ethical Concerns:**

["NO or VERY MINOR ethics concerns only"]

**Final Justification:**

No change on the review.

**Limitations:**

yes

**Paper Formatting Concerns:**

No specific concerns about paper formatting.

**Quality:**

2

**Strengths And Weaknesses:**

### Strengths
- **Unified Feedback Integration:** Seamlessly combines demonstrations, preferences, and rankings into a single reward-learning framework.
- **Robustness under Scarcity:** Demonstrates superior performance in low-data regimes and when feedback is heterogeneous.

### Weaknesses
- **Simple Evaluation Domains:** Tasks are toy-scale with well-known ground-truth rewards; it is unclear whether LEOPARD scales to complex environments that require intricate reward engineering.
- **Lack of Ablation on Feedback Types:** No quantitative analysis of each feedback modality’s efficiency or relative contribution to the learned reward.
- **Fragment Independence Assumption:** Assumes trajectory fragments are independent, which may break down in long-horizon tasks (e.g., AntMaze) where small fragments fail to capture causality and long-term dependencies.
- **Strict Ordering Assumption:** Imposes a rigid ordering among negative demonstrations, agent-generated fragments, and positive demonstrations; since agent behavior quality evolves, injecting mixed orderings could introduce noise into the reward learning process.

---

> ### Author Rebuttal · Authors · 2025-07-30
>
> We thank the reviewer for their comments. We are particularly grateful to the reviewer for highlighting our framework’s robustness and seamless integration of multiple feedback types. We completely understand the reservations held by the reviewer; below, we provide responses to the reviewers’ questions and concerns. We hope that these are sufficient for the reviewer to raise their score.
>
> **(W.1) Evaluations.**
> We thank the reviewer for this point, and acknowledge that our evaluation could have been more extensive. We deliberately designed it to test the core hypothesis across diverse control paradigms. The four environments span discrete/continuous control, different state representations, and varying reward sparsity. This allows us to demonstrate clear, statistically significant improvements that can be expected to replicate on other tasks. While broader evaluation would be valuable, our current results establish clear superiority over existing methods. We are confident LEOPARD will scale well to other tasks, as it can efficiently learn from lots of feedback data in linear time, whilst also being able to handle environments with high dimensional state and action spaces.
>
> **(W.2) Feedback type ablation.**
> We perform an experiment to measure the impact of different feedback types in section 4.3. We acknowledge a more detailed analysis could be performed, but the results presented give reasonable intuition on how each feedback modality affects the learnt reward.
>
> **(W.3) Independence assumption.**
> The reviewer is correct that strict independence is a rare condition, however, we only assume independence conditioned on the underlying true reward function. This is a necessary and common assumption that makes the problem tractable to solve. Prior work shows that a better assumption is that preferences arise from regret, but that the assumption we make works well in practice and leads to a well-shaped reward function [1].
>
> **(W.4) Strict ordering assumption.**
> Whilst we assume a strict ordering, we also assume Boltzmann-Rationality, which allows for perturbations from the ordering assumption to be tolerated if they allow the learnt reward function to better capture the rest of the data. Thus in practice this is not an issue, and empirically we often see agent behaviour far exceeding the quality of the demonstrations when preferences are also being used, as shown for example by comparing figure 4a with figure 2a.
>
> **(Q.1) Multi-objective trade-offs.**
> LEOPARD is built upon RRPO, which maximises the likelihood of the feedback data under a Boltzmann-Rational model. This grounding in well understood and well established models gives us confidence that as the task complexity increases these trade-offs will be handled robustly. We also do not expect there to be significant conflict between the feedback types that might induce the need for strong trade-offs to be made. Given the demonstrations and preferences are being generated with reference to the same underlying ground truth reward (implicitly in the case of real human feedback), we expect that fitting one well is strongly correlated with fitting the other well. If the reviewer disagrees with this intuition, please could they clarify what sort of specific trade-offs they are worried might be mishandled?
>
> **(Q.2) Feedback budget selection.**
> When developing LEOPARD and evaluating it, we had to choose the right amount of feedback and agent rollout steps in order to get meaningful results. Too few of either would make it impossible to get a good ground truth reward, regardless of the qualities of the training algorithm. Likewise, if there was an overwhelming amount of feedback and a huge interaction budget for the agent itself, learning would be too easy, and many algorithms would provide good final performance. Thus, we chose an amount of feedback and interaction budget such that LEOPARD performed moderately well, but not excellently, so that methods which were better or worse than it could be easily identified. We then tested the baselines under the same conditions presenting these results in the paper. We direct the reviewer to section 4.1, particularly the final paragraph, and Appendix B for more details on our experimental setup.
>
> **(Q.3) Guaranteed balanced mixing.**
> Whilst the optimal mix of feedback types does vary amongst domains, as noted in section 4.3, using positive demonstrations and preferences is the most reliable way to obtain solid performance across a wide variety of domains. We expect practitioners to use all feedback they have available to achieve the best performance, and LEOPARD supports this. As for domain-specific manual tuning, it does not need to be performed. With two exceptions, all hyperparameters were consistent across all domains. This included the number of iterations, the network architecture, the reward model training procedure, and the hyperparameters of the RL algorithm. The two exceptions were a slightly smaller preference fragment length for Cliffwalking, and whether PPO or SAC was used, which was based on whether the action space was discrete or continuous. We believe the high performance of our method across these diverse domains given fixed hyperparameters is further evidence our method will generalise well.
>
> **(Q.4) Baseline comparison.**
> We have not tested against Preference Aided Imitation Learning (PAIL), though we acknowledge this would be an interesting and insightful comparison. Our reasoning for not including it was that it is intended for learning from suboptimal demonstrations with an unknown ordering via preferences over these demonstrations, rather than demonstrations with a known ordering combined with additional preferences over agent behaviour. Furthermore, it is not able to learn from intentionally negative or failed demonstrations. We realise that a method incorporating the insights and advantages from both LEOPARD and PAIL might be very effective and interesting, but this is outside the scope of our paper.
>
> We hope that the reviewer is satisfied by our response. We kindly ask that the reviewer raises their score in light of our responses, or otherwise specify what further changes they would like to see to the paper which would lead them to raise their score. Once again, we thank the reviewer for their engagement with our manuscript and for their thoughtful commentary and feedback.
>
> [1] Knox, W. B., Hatgis-Kessell, S., Adalgeirsson, S. O., Booth, S., Dragan, A., Stone, P., & Niekum, S. (2024). Learning Optimal Advantage from Preferences and Mistaking It for Reward. Proceedings of the AAAI Conference on Artificial Intelligence, 38(9), 10066-10073. https://doi.org/10.1609/aaai.v38i9.28870

---

> > ### Comment · Reviewer_VZqx · 2025-08-05
> >
> > Thanks for your rebuttal. Since my main concern about the lack of evaluation remains unresolved, I would like to keep the original score. I instead recommend the authors conduct additional experiments in another simulation environment besides Gymnasium.

---

> > > ### Author Response · Authors · 2025-08-08
> > >
> > > We thank the reviewer for their engagement with our rebuttal and for their suggestions.
> > >
> > > We understand that the number of evaluation environments in the manuscript are insufficient to convince the reviewer of the efficacy of our method. To aid us in improving our work for future submissions, we kindly ask the reviewer what sorts of environments they think would be particularly important for us to evaluate our method on?

---

### Official Review · Reviewer_TpHB · 2025-07-02

**Clarity:** 2
**Significance:** 2
**Originality:** 2
**Rating:** 2
**Confidence:** 4

**Summary:**

The paper attempts to learn from a combination of preferences and mixed demonstrations (positive and negative) by modeling human feedback using theory of reward rational choice ordering. The paper begins with going over modeling human feedback using Boltzmann Rational Choices and using this to learn a reward model. Then the paper extends this to guide policy search by using several forms of feedback such as preferences and demonstrations (positive and negative). The paper follows this up with benchmarking the proposed method and baselines (imitation learning with preferences, and deep IRL then RLHF) on four environments from gymnasium.

**Questions:**

1. Could the authors expand upon their framework to study how the proposed framework converges with transitive, intransitive preferences? Does it recover Nash when we have intransitive preferences?
2. How does the framework scale up with simple LLM experiments even for a single task such as helpfulness vs harmlessness?
3. What happens if we employ a simple RLHF adaptation where we use pairwise RMs for preferences and pointwise for positive and negative demonstrations and utilize a combination of these reward models for policy search?

**Ethical Concerns:**

["NO or VERY MINOR ethics concerns only"]

**Final Justification:**

[I submitted mandatory acknowledgement and responded to the authors during the discussion period.]

The authors faithfully engaged to address feedback during the discussion period, and towards this I am grateful to the authors. However, as I mentioned during the discussion period, the paper presents interesting ideas that need more work to be operationalized for scenarios that occur often in practical settings. With this in mind, I will retain my score as is and thank the authors for their work.

**Quality:**

2

**Strengths And Weaknesses:**

Strengths:
- Very interesting and well motivated problem, has ample scope to derive new algorithms

Weaknesses:
- The methodology section appears to be fairly thin in terms of details - in particular, the writing when describing the proposed framework just appears to be described using a few lines of math and not much explanations (see lines 200-210).
- Limited analysis on when the proposed framework works, and what are the failure modes
- The experimental simulations with gymnasium shows promise in the realms of continuous control but again this is rather limited when it comes to judging the framework's promise.

---

> ### Author Rebuttal · Authors · 2025-07-30
>
> We thank the reviewer for their comments. We are particularly grateful to the reviewer for highlighting the motivation behind our new methods. We completely understand the reservations held by the reviewer; below, we detail the changes we have made to the manuscript and provide responses to the reviewers’ questions and concerns. We hope that these are sufficient for the reviewer to raise their score.
>
> **(W.1) Exposition of method.**
> We thank the reviewer for highlighting that the main explanation of our method is thin on details. We have updated the manuscript to make this clearer. We note that more comprehensive details of the method and evaluation setups are provided in Appendices A and B respectively, including the full algorithm and notes on subtle implementation details.
>
> **(W.2) Analysis of when the framework fails.**
> We acknowledge that a full analysis of when and how the method fails would be interesting, and thank the reviewer for highlighting this. On our set of evaluations we did not observe any notable trends of failure that felt significant. If there are specific experiments or analyses the reviewer had in mind, we’d be more than happy to investigate them. On the theoretical side, the method is solid, and designed to operate in very general contexts. Beyond Boltzmann-Rationality, which is a common and well justified assumption, we intentionally do not make any strong assumptions about the nature of the problem or the structure of the reward signal. Thus, there are no specific contexts where we’d expect our method to fail when other Boltzmann-Rational or deep learning based methods would succeed.
>
> **(W.3) Evaluation breadth.**
> We thank the reviewer for this point, and acknowledge that our evaluation could have been more extensive. We deliberately designed it to test the core hypothesis across diverse control paradigms. The four environments span discrete/continuous control, different state representations, and varying reward sparsity. This allows us to demonstrate clear, statistically significant improvements that can be expected to replicate on other tasks. While broader evaluation would be valuable, our current results establish clear superiority over existing methods.
>
> **(Q.1) Convergence with intransitive preferences.**
> The reviewer makes an excellent point. Whilst we have proven and empirically demonstrated the performance of our algorithm under ideal conditions, we have not explored the theory of how it would behave when faced with intransitive preferences. As such, we have updated our manuscript to explore this in an Appendix chapter. The headline result, which also holds for RLHF, is that given $K$ preferences of $A<B$ and $N$ preferences of $B<A$, the loss is minimised when $R(A) - R(B) = \log(N/K)$, which we feel is a very reasonable and moderate behaviour when faced with this situation. This extends to settings beyond pairwise preferences, but it becomes slightly more complex. Furthermore, the loss from intransitive preferences can be subtracted from the overall loss, and the theorem concerning loss bounds and representing the rest of the data still applies. Finally, we note that in our evaluations preferences were generated stochastically, and so some intransitivity might have occurred. However, it seems this did not significantly negatively affect LEOPARD.
>
> **(Q.2) LLM experiments.**
> We thank the reviewer for highlighting that testing how LEOPARD performs at training LLMs would be very interesting. In principle the framework should scale gracefully to be able to be tested on LLMs as it has been formulated to be domain agnostic and data efficient. As shown by its performance on other tasks, it should only take a small number of iterations to obtain good performance. The main difficulty here is that LLMs are typically optimised with RLHF in an offline manner—the reward model is not re-trained once it has been fit to a dataset of preferences. LEOPARD however would necessitate its re-training in order to absorb the information present in the demonstrations as the LLM explores various styles. We imagine that taking a reward model fit to a static preference dataset, and then further fine-tuning this along with the policy LLM with LEOPARD would yield great results. However, testing on LLMs involves significant research engineering effort and computation costs, and for our initial set of experiments we wanted to motivate and validate the performance of LEOPARD on more classic problems. We plan to test our algorithm in the LLM setting as future work.
>
> **(Q.3) Reward model combination.**
> This is an interesting idea, and one we considered ourselves. Our main motivation for not exploring it further was three-fold. Firstly, without additional regularisation it might be difficult to get the reward scales of the separate reward models to be comparable, and thus one may dominate during policy optimisation, leaving lots of useful feedback under utilised. This is not an issue with RRPO, as it casts the problem as likelihood maximisation, providing a unified framing for all feedback types. Secondly, separate reward models may fail to capture reward structure that is only inferrable from multiple sources of information. As a slightly contrived but instructive example, consider a 2D real-valued state space with two reward functions, $R_1$ and $R_2$, both trained on a different preference. $R_1$ is trained on $([1, 1] > [-1, 1])$ and implements $R_1([x, y]) = x$. $R_2$ is trained on $([-1, -1] > [1, -1])$. $R_1$ implements $R_2([x, y]) = -x$. Now consider $R_3$, which is trained on both preferences and implements $R_3([x,y]) = xy$. We observe that $R_1+R_2 = 0 \neq R_3$. The reward functions have learnt something very different! In general this could happen if there was some non-linear aspect to the reward which no single reward model has enough data to identify, that could be identified from the combination of some sources of data. Finally, we are interested in a method that will work when combining many feedback types together, not just preferences and demonstrations. It may be that for some of these feedback types there is very little data present. For RRPO, this is not an issue, the small amount of data has a correspondingly small effect on the loss and the learnt reward. However, if learning separate reward models for each feedback type, a model learning from only a small amount of data might pick up on a spurious pattern and generalise it, providing potentially large, incorrect reward predictions. This would greatly degrade the efficacy of this approach.
>
> We hope that the reviewer is satisfied by our response and the corresponding changes we have made to the manuscript. We kindly ask that the reviewer raises their score in light of our responses, or otherwise specify what further changes they would like to see to the paper which would lead them to raise their score. Once again, we thank the reviewer for their engagement with our manuscript and for their thoughtful commentary and feedback.

---

> > ### Comment · Reviewer_TpHB · 2025-08-05
> > **Response to author rebuttal**
> >
> > Thank you to the authors for their response. While the authors take a step towards addressing some of the shortcomings, I still believe the paper requires sufficiently additional effort (and requires going through a review process) to validate its contributions relative to the literature at hand. With this in mind, I plan to retain my score as is and thank the authors for their efforts in trying to strengthen their work.

---

> > > ### Author Response · Authors · 2025-08-08
> > >
> > > We thank the reviewer for engaging with our rebuttal and for recognising the steps we've taken to address some of the shortcomings of our work.

---

### Official Review · Reviewer_XNHd · 2025-07-03

**Clarity:** 3
**Significance:** 2
**Originality:** 2
**Rating:** 3
**Confidence:** 3

**Summary:**

This paper proposes LEOPARD, a reward-learning framework designed for mixed human feedback datasets, which include comparisons as well as positive and negative demonstrations. Motivated by reward-rational choice (RRC), the paper introduces the RRPO loss, which maximizes the log-likelihood of selecting a given trajectory over all trajectories ranked lower in the provided partial orderings. Empirical evaluations demonstrate that LEOPARD outperforms two baseline methods, DeepIRL + RLHF and AILP, across various environments. The results also confirm that incorporating multiple types of feedback data leads to better performance compared to relying on a single type.

**Questions:**

- In Section 4.3, it is common for reward-learning algorithms to achieve better performance by combining multiple types of feedback data, as doing so allows reward functions to be learned across a broader state-action space, thereby mitigating reward ambiguity or degeneracy. Therefore, I believe this result is not specifically an advantage of LEOPARD but rather a general phenomenon. Could the authors comment on this?
- It is unclear why using the RRPO loss would be better than simply aggregating the Bradley-Terry loss across all partial orderings. Could the authors provide more intuition or explanation regarding this?
- In Figure 2, the performance of the two baseline methods seems too poor in some cases, showing little improvement from the arly training stages. Could the authors clarify why this occurs in the baselines?

**Ethical Concerns:**

["NO or VERY MINOR ethics concerns only"]

**Final Justification:**

While the proposed framework is interesting and promising, I think the evaluation of the algorithm and the presented empirical evidence are limited to fully support the authors’ claims. I will keep my score as it is.

**Limitations:**

Yes.

**Paper Formatting Concerns:**

No formatting concerns found.

**Quality:**

2

**Strengths And Weaknesses:**

**Strengths:**
- The proposed framework is easy to implement, scalable, and generalizable across diverse environments and tasks.
- The proposed method is generally well-motivated and clearly presented.

**Weaknesses:**
- The baseline methods used for comparison in experiments are not explicitly designed to address multiple feedback data types. It would have been more interesting to see the comparisons with algorithms specifically developed for this setting.
- The provided theoretical analysis covers only representation properties and does not address generalization or statistical guarantees.

---

> ### Author Rebuttal · Authors · 2025-07-30
>
> We thank the reviewer for their comments. We are particularly grateful to the reviewer for highlighting the ease of implementation, scalability, and generality of our method. We completely understand the reservations held by the reviewer; below, we detail the changes we have made to the manuscript and provide responses to the reviewers’ questions and concerns. We hope that these are sufficient for the reviewer to raise their score.
>
> **(W.1) Baseline comparisons.**
> We thank the reviewer for raising this point and acknowledge that testing against a broader range of baselines would help strengthen our evaluation. The MaxEnt IRL followed by RLHF baseline was chosen as we feel it represents a simple and standard approach to learning from both demonstration and preference data when no purpose-built algorithms are being employed. Adversarial Imitation Learning with Preferences (AILP) was chosen as it is a recent combined preference and demonstration learning algorithm that is very effective and specifically designed to learn from both preferences and demonstrations. If the reviewer had other specific algorithms in mind, we’d be happy to test them to improve future iterations of our paper.
>
> **(W.2) Theoretical analysis.**
> We thank the reviewer for highlighting that the theoretical analysis is currently limited, and for expressing a desire to see the theoretical properties of RRPO explored further. We acknowledge that results on generalisation or statistical guarantees would be interesting. We have updated the manuscript to include a new analysis of the case where the preferences are intransitive, as our current theoretical result assumes this to not be the case.
>
> **(Q.1) Performance when learning from multiple feedback types.**
> We thank the reviewer for highlighting an area of the paper where the intentions of our evaluations were unclear. We agree that in general methods that can learn from many feedback sources would likely improve in performance as the types of feedback used increases, and that this is not unique to LEOPARD. We intended our results in section 4.3 to explore the strength and nature of this effect for our algorithm in particular. We rely upon the results of section 4.2 to support the claim that LEOPARD has superior performance to other mixed feedback reward learning methods. We have updated the manuscript to make the intentions of our experiments and results clearer.
>
> **(Q.2) RRPO loss versus aggregated Bradley-Terry loss.**
> This is an interesting question, and we acknowledge that the novelty of the provided loss function is somewhat unintuitive. The loss function we derived has three major advantages when compared to the procedure as described by the reviewer. The first is that by encoding the demonstration data into a partial ordering, rather than a set of binary choices, both negative demonstrations and rankings over the demonstrations themselves can be directly incorporated without ad-hoc alterations to the loss or algorithm. The second is sample efficiency. In the (positive) demonstration only case, pairwise sampling preferences via the demonstrations might have a similar result to the RRPO demonstration loss in the limit of infinite samples, but we’d expect it to converge much slower. This is because the RRPO loss can efficiently encode that a batch of positive demonstrations are all superior to a batch of agent trajectories, and the optimisation pressure can be diverted to pushing down the best-rated agent trajectory and pushing up the worst-rated demonstration. For $K$ demonstrations and $N$ agent trajectories, what RRPO can achieve in a single batch of size $K+N$ would take $K \times N$ pairwise preferences, thus making it far more efficient. The third major advantage is that, for rankings over demonstrations, these rankings can also be learnt much more efficiently versus sampling preferences from them. Consider four demonstrations of increasing quality, $A<B<C<D$. If the sampled preferences are for example $A<B$, $B<C$, $C<D$, and the reward model initially assigns each demonstration equal reward, then the upward and downward push on B from $A<B$ and $B<C$ both cancel out. The same happens for $C$, and after one optimisation step the reward model will give approximately the following: $A<B~=C<D$. For RRPO however, the full ordering is learnt from, and a single step will instantly yield the correct $A<B<C<D$ ordering. This is another case where the limit of infinite preference sampling will eventually yield the correct solution, but the clever loss of RRPO computes the best direction to move the weights in at every step.
>
> **(Q.3) Poor baseline performance.**
> We were also somewhat initially surprised by the poor performance of the baseline methods, and were worried their implementations were incorrect. However, considering the broader context of our evaluations, as well as our other results, makes the picture much clearer. When developing LEOPARD and evaluating it, we had to choose the right amount of feedback and agent rollout steps in order to get meaningful results. Too few of either would make it impossible to get a good ground truth reward, regardless of the qualities of the training algorithm. Likewise, if there was an overwhelming amount of feedback and a huge interaction budget for the agent itself, learning would be too easy, and many algorithms would provide good final performance. Thus, we chose an amount of feedback and interaction budget such that LEOPARD performed moderately well, but not excellently, so that methods which were better or worse than it could be easily identified. We then tested the baselines under the same conditions presenting these results in the paper. We believe that given larger amounts of feedback, and more interaction time with the agent, the other baseline methods would do much better (and LEOPARD would further improve). In fact, this prediction is already verified by the set of experiments comparing demonstration-only learning, visualised in Appendix C, figure 6. Here we see many of the baselines doing much better than their counterparts figure 2, where they have only half the amount of demonstration data (though can additionally utilise preferences). We note that LEOPARD is still superior in these settings, particularly for the more difficult tasks. We acknowledge that many more sweeps over a large range of feedback mixtures with comprehensive results presented for each algorithm would help clarify our results and the relative performance of our method. However, we felt given the strength of our results for the feedback mixtures we tested on, this was not necessary to strengthen our claims.
>
> We hope that the reviewer is satisfied by our response and the corresponding changes we have made to the manuscript. We kindly ask that the reviewer raises their score in light of our responses, or otherwise specify what further changes they would like to see to the paper which would lead them to raise their score. Once again, we thank the reviewer for their engagement with our manuscript and for their thoughtful commentary and feedback.

---

> > ### Comment · Reviewer_XNHd · 2025-08-04
> >
> > Thank you for the clarification. While I mostly agree with the authors’ responses and find the proposed framework promising, the current draft lacks sufficient clarity, particularly in the evaluation and the discussion of limitations, for me to recommend acceptance. I will maintain my original score at this point.

---

> ### Author Response · Authors · 2025-08-05
>
> We thank the reviewer for their comment and their engagement with our rebuttal.
>
> It would be helpful if the reviewer could be more specific about how clarity is lacking, or could be improved in those sections, as we may be able to address this problem before the discussion period is over and improve the manuscript.
>
> Finally, if the reviewer is happy with our responses to their questions and identified weaknesses, please could they clarify what information, changes, or improvements they would require in order to raise their score.

---

> > ### Comment · Reviewer_XNHd · 2025-08-08
> >
> > To clarify, I am not entirely sure whether the empirical evidence and discussion in Sections 4 and 5 are sufficiently clear to support the advantages of the algorithm.
> >
> > The presented training curves show the performance of the learned policies, but they do not provide direct evidence for each specific component or claim made about the algorithm, which makes it difficult for me to interpret the implications.
> >
> > For example, if the authors believe that the RRPO loss is more sample efficient in aligning with the trajectory ranking data than simply combining pairwise losses, they could compare the alignment error directly. In Section 4.3, the authors could compare the performance gains against different baseline algorithms to show when and why LEOPARD is particularly good at utilizing multiple data types.
> >
> > Could the authors provide further comment on this? I would be willing to reconsider my score if my concern is misplaced.

---

> > > ### Author Response · Authors · 2025-08-08
> > >
> > > Thank you for the clarification and for engaging with our comment.
> > >
> > > With regards to sample efficiency, this is evidenced by Leopard often achieving a much higher reward than the baselines in few iterations, and thus few agent rollout steps. (Note there are a fixed number of agent rollout steps per iteration of the algorithm.)
> > >
> > > In Section 4.3, we do not compare to any baseline methods as we are not aware of any other method that is flexible enough to learn from preferences, positive and negative demonstrations, and demonstration rankings, simultaneously. One could construct an amalgamation of different methods and reward models or pairwise losses for each of the different feedback types, but this would be non-trivial in design. Many design choices would need to be made in the aggregation of these reward models or what data pairwise comparisons were constructed from. We believe that constructing such a method, especially a variant that is well theoretically motivated, would in itself represent a new feedback learning algorithm. Thus we did not pursue it and instead focused on our own, simpler method.

---

### Official Review · Reviewer_c3ZL · 2025-07-07

**Clarity:** 4
**Significance:** 2
**Originality:** 3
**Rating:** 4
**Confidence:** 4

**Summary:**

This work addresses the problem of learning policies from a combination of both preference-based feedback and task demonstrations.  It presents LEOPARD, an approach that integrates demonstrations and preferences by converting both into a partial preference ordering over trajectory fragments, and then using a loss inspired by the RRiC framework to learn a reward function maximizing the likelihood of this partial ordering.  There seem to be two key ideas behind this work:
1. Interpreting demonstrations as "implicit" preferences for the demonstrated trajectories over a collection of alternative trajectories (either AI generated or provided by yhe user).
2. In contrast to existing RLHF methods, the training loss employed by LEOPARD does not assume that individual preference orderings are sampled independently, but instead considers the joint probability of the entire set of preferences.

Experiments in a set of standard RL benchmark tasks (including the MuJoCo Ant and Half-Cheetah) environments, demonstrate that LEOPARD significantly outperforms two baseline approaches to combining preferences and feedback.

**Questions:**

1. How would LEOPARD be expected to compare against a baseline that translates demonstrations to preferences in the same way, but uses a standard RLHF loss (which treats each preference as being independent)?  That is, how important is the novel loss function (Equation 4) itself?
2. Was there a specific reason why behavioral cloning on the demonstration trajectories was not used as a baseline, with AILP and RLHF being the only baselines?
3. In Figures 2 and 3, what does one "iteration" correspond to?

**Ethical Concerns:**

["NO or VERY MINOR ethics concerns only"]

**Final Justification:**

The rebuttals addressed some of my concerns, but I still believe the paper would be more convincing if LEOPARD were compared against the additional baselines mentioned in W3 and Q1.  I therefore believe my original score was appropriate.

**Limitations:**

Yes

**Paper Formatting Concerns:**

N / A

**Quality:**

3

**Strengths And Weaknesses:**

**Strengths:**
1. The proposed training loss is novel, and may be able to capture information conveyed through the choice of a specific *set* of preferences (or demonstrations) that a loss based on the Bradley-Terry model might ignore.
2. The experimental results show that LEOPARD outperforms both baseline methods by a wide margin, at least in the two MuJoCo tasks evaluated.

**Weaknesses:**
1. While this work builds on the RRiC model, it seesm the loss used in LEOPARD does not incorporate one of the key insights of RRiC, namely, the idea that demonstrations (and preferences feedback) are selected by the human teacher with the aim of maximizing the expected reward under the distribution over AI trajectories that this demonstration induces.  The RRPO loss (4) still assumes that choices are made based on the advantage of the preferred trajectory over all other trajectories in the choice set.
2. While RRPO combines preferences and demonstrations, it is not clear whether it fully incoprorates demonstration data.  The reward model may focus on specific components of the trajectory, and ignore differences between the final policy learned and the action choices observed in the demonstrations themselves.  As demonstrations can often communicate far more information that preferences, this would seem to put the method at a disadvantage to direct imitation learning methods.
3. The experimental evaluation could have been more extensive, for example, evaluating across more (and more challenging) tasks, and comparing against additional baselines (in particular behavioral cloning from the positive demonstrations alone, and DPO for preferences alone).

---

> ### Author Rebuttal · Authors · 2025-07-30
>
> We thank the reviewer for their comments. We are particularly grateful to the reviewer for highlighting the novelty of our method and its strong performance versus the baselines. We completely understand the reservations held by the reviewer; below, we detail the changes we have made to the manuscript and provide responses to the reviewers’ questions and concerns. We hope that these are sufficient for the reviewer to raise their score.
>
> **(W.1) Demonstration selection assumption.**
> Whilst it’s true that the assumption of RRC that demonstrations are chosen to maximise expected reward of the learnt policy is a key insight, RRPO replaces this with an alternative assumption that demonstrations are chosen to be favoured to agent trajectories during training. We do this as the RRC formulation leads to a demonstration loss that is intractable to directly compute, and requires approximating, whereas the RRPO loss is directly computable. We found that this avoids problems such as the surrogate demonstration loss becoming unbounded from below, and thus dominating the overall loss leading to preferences being ignored, or preventing the use of early stopping techniques to optimally train the reward network each iteration without overfitting.
>
> **(W.2) Incorporation of demonstration data.**
> We thank the reviewer for raising this interesting point. We agree that demonstrations contain lots of information, and that capturing all this information is critical to high performance when learning from multiple sources of feedback. Indeed, this was one of our motivations for developing LEOPARD. We can reassure the reviewer that LEOPARD fully incorporates demonstration data, to the extent it does not conflict with other feedback types. The reward model is essentially being trained to differentiate the agent’s attempts and the demonstrations, so that it can provide the latter with higher reward to better fit the RRPO model. It can exploit any differences between them to assign differing reward scores, including which actions are taken in which states. This is possible as the whole demonstration is used, instead of merely being sampled from. If any differences exist, the reward model will offer higher reward for the actions and states found in the demonstrations, and lower reward for those found in the agent’s attempts. Thus the agent can then increase its score by taking these actions that are being rewarded. We see that as reward model and policy training are alternated, the policy state-action distribution must converge to exactly match the demonstrations if there is no other feedback data and sufficient exploration (as any divergence can be exploited by the learnt reward function). When preferences are incorporated, the exact point of convergence will change as there is more information of the underlying ground truth reward, but all the demonstrations are still being made full use of.
>
> **(W.3) Evaluation breadth.**
> We thank the reviewer for this point, and acknowledge that our evaluation could have been more extensive. We deliberately designed it to test the core hypothesis across diverse control paradigms. The four environments span discrete/continuous control, different state representations, and varying reward sparsity. This allows us to demonstrate clear, statistically significant improvements that can be expected to replicate on other tasks. While broader evaluation would be valuable, our current results establish clear superiority over existing methods.
>
> **(Q.1) Comparison against preference-based demonstration learning.**
> This is an interesting question, and we acknowledge that the novelty of the provided loss function is somewhat unintuitive. The loss function we derived has three major advantages when compared to the procedure as described by the reviewer. The first is that by encoding the demonstration data into a partial ordering, rather than a set of binary choices, both negative demonstrations and rankings over the demonstrations themselves can be directly incorporated without ad-hoc alterations to the loss or algorithm. The second is sample efficiency. In the (positive) demonstration only case, pairwise sampling preferences via the demonstrations might have a similar result to the RRPO demonstration loss in the limit of infinite samples, but we’d expect it to converge much slower. This is because the RRPO loss can efficiently encode that a batch of positive demonstrations are all superior to a batch of agent trajectories, and the optimisation pressure can be diverted to pushing down the best-rated agent trajectory and pushing up the worst-rated demonstration. For $K$ demonstrations and $N$ agent trajectories, what RRPO can achieve in a single batch of size $K+N$ would take $K \times N$ pairwise preferences, thus making it far more efficient. The third major advantage is that, for rankings over demonstrations, these rankings can also be learnt much more efficiently versus sampling preferences from them. Consider four demonstrations of increasing quality, $A<B<C<D$. If the sampled preferences are for example $A<B$, $B<C$, $C<D$, and the reward model initially assigns each demonstration equal reward, then the upward and downward push on the reward of $B$ from $A<B$ and $B<C$ both cancel out. The same happens for $C$, and after one optimisation step the reward model will give approximately the following: $A<B~=C<D$. For RRPO however, the full ordering is learnt from, and a single step will instantly yield the correct $A<B<C<D$ ordering. This is another case where the limit of infinite preference sampling will eventually yield the correct solution, but the clever loss of RRPO computes the best direction to move the weights in at every step.
>
> **(Q.2) Baselines.**
> We acknowledge that additional baselines to compare against would be interesting, however we focussed our baselines to the best methods that were intended, or used in practice, for learning from both preference and demonstration data. We note that specifically for behavioural cloning, we feel that based on prior work it would perform much worse than Maximum Entropy Deep IRL, as behavioural cloning struggles in complex problems with infinite high dimensional state and action spaces. We did not test on DPO as, unlike the language modelling setting, the tasks we test on are not reducible to contextual bandits. This reduction is required to make the core theoretical result of the DPO paper valid, and the DPO method is directly derived from this [1]. However, we thank the reviewer for these ideas, and note that they would be very interesting baselines to test against in future work when we evaluate LEOPARD in the language modelling setting, as behavioural cloning and DPO are highly performant in this domain.
>
> **(Q.3) The meaning of ‘iteration’.**
> Thank you for highlighting that this is a little ambiguous, and that the current explanation in the figure captions is less than ideal. We have altered the manuscript to make this clearer. The LEOPARD algorithm consists of alternating between training the reward model and training the agent. We use ‘iteration’ to refer to one complete cycle of this process, i.e. one iteration covers training the reward model on the current set of data to convergence, and then training the agent on the new reward signal for a fixed number of rollout steps. For further clarification and detail on the exact training procedure, please refer to algorithm 1 in Appendix A.
>
> We hope that the reviewer is satisfied by our responses and the corresponding changes we have made to the manuscript. We kindly ask that the reviewer raises their score in light of our responses, or otherwise specify what further changes they would like to see to the paper which would lead them to raise their score. Once again, we thank the reviewer for their engagement with our manuscript and for their thoughtful commentary and feedback.
>
> [1] Rafailov, R., Sharma, A., Mitchell, E., Manning, C.D., Ermon, S. and Finn, C., 2023. Direct preference optimization: Your language model is secretly a reward model. Advances in neural information processing systems, 36, pp.53728-53741.

---

> > ### Comment · Reviewer_c3ZL · 2025-08-05
> > **Response to Rebuttals**
> >
> > Regarding W1, I agree that it would be impractical to explicitly compute and optimise the RRiC likelihood.  It would be worth explaining the distinction between this method and RRiC more clearly, however.
> >
> > You may be right that LEOPARD would outperform some of the simpler baselines I mentioned, but you would really need experimental results to support this argument.

---

> > > ### Author Response · Authors · 2025-08-08
> > >
> > > Thank you for engaging with our rebuttal.
> > >
> > > We agree that this distinction could be made clearer, and we thank the reviewer for this suggestion.
> > >
> > > We also agree that experimental results are important, and are required for one to be confident in claims about one method vs another. However, for this problem there are many different methods one could employ, and many of these seem fairly simple or straightforward. In particular, in Appendix E we explore two other simple-seeming baselines derived from RRC and show how they are theoretically deficient. They also performed poorly in early experiments. Because of the wide range of potential baselines and alternative methods, we decided to keep our evaluation focused to those we thought would perform the strongest, and thus be the best alternative compared to Leopard. This helped us avoid spending significant amounts of time and money on the compute required to exhaustively evaluate many baselines.

---

### Decision · Program_Chairs · 2025-09-17

**Decision:**

Reject

**Comment:**

The authors study the problem of learning policies via reinforcement learning from multiple types of human feedback including pairwise preferences, positive demonstrations, negative demonstrations, and partial demonstration rankings. Specifically, they introduce a general mathematical framework that frames these human feedback types as reward-rational partial orderings (RRPO), building on the reward-rational choice (RRC) formulating by simplifying representing feedback as distributions over trajectories (which is difficult to optimize due to combinatorial considerations) to representing feedback as partial orderings (which is clearly easier to optimize). The specific instantiation presented, LEOPARD (Learning Estimated Objectives from Preferences and Ranked Demonstrations), integrates feedback from demonstrations and preferences by converting both to partial preference orderings over trajectory fragments -- which is used to learn a reward function based on the partial orderings. The key insights are to represent demonstrations as partial preferences (thus simplifying optimizing over full demonstration trajectories) while having the training loss consider the joint probability of the set of preferences (includes of independent samples as in most RLHF formulations). Experiments are conducted on multiple benchmark environments from the Gymnasium test suite, significantly outperforming two baselines that combine different feedback types (Adversarial Imitation Learning with Preferences (AILP) and DeepIRL+Preference fine-tuning with RLHF) -- showing that learning a reward function and corresponding policy can benefit from multiple feedback types.

Strengths of this work identified by the reviewers include:
- The proposed training loss is novel and has conceptually appealing advantages over the widely-used Bradley-Terry model for learning preferences. The implementation is relatively straight-forward and scales/generalizes to many settings
- The experimental results are positive and convincingly demonstrate the key claims of LEOPARD.

Weaknesses of this work identified by the reviewers include:
- There isn't a sufficiently precise contrast of the advantages and disadvantages of RRC vs. RRPO (and more specifically LEOPARD). While this was sufficiently addressed during the rebuttal period (in my opinion; e.g., Q.1 for c3ZL response), there is some remaining work to do this in the paper (possibly even including targeted experiments to do well).
- The experimental results are narrowly focused on Gymnasium tasks to show the strengths of LEOPARD as compared to the nearest (most natural) benchmarks. However, the consensus amongst the reviewers was that more tasks and more baselines are necessary to make the results convincing. While I am a bit more convinced than the reviewers (even after rebuttal), I do agree that more extensive experiments would make for a stronger paper.
- RRPO is defined over many (actually, probably all practical) feedback types. However, LEOPARD (and thus the experiments) are limited to (mostly positive) demonstrations and preferences and the results presented in Section 4.3 & Appendix C lack thoroughness and sufficient discussion.
- There is limited theoretical analysis (which is fine, but the reviewers felt could be stronger, even after rebuttal).
- While targeted for future work, the natural question is going to be the language model setting -- and it would clearly be interesting to see even some preliminary experiments in this direction (e.g., post-training a mid-sized LLM).

Based on the consensus amongst reviewers, even after rebuttal, there was a desire for more complex experiments, more baseline system configurations, and more interpretation of the empirical results. As stated previously, I am a bit more convinced than the reviewers regarding the experiments being sufficient in validating the key ideas, but also concede that more complex experiments (including possibly LLM post-training) would make for a much stronger paper. Regarding clarification questions and better contrast with RRC, this was sufficiently addressed in rebuttal but still needs to be better integrated in the manuscript. However, even after discussion, the reviewers remained adamant that more experiments are needed to sufficiently support the key claims of the paper.